# Intrinsic room-temperature ferromagnetism in a two-dimensional semiconducting metal-organic framework

Sihua Feng[1], Hengli Duan [1] ✉, Hao Tan [1], Fengchun Hu[1], Chaocheng Liu[1], Yao Wang[1], Zhi Li[1], Liang Cai [1], Yuyang Cao[1], Chao Wang [1] ✉, Zeming Qi[1], Li Song [1], Xuguang Liu[2], Zhihu Sun [1] & Wensheng Yan [1] ✉

The development of two-dimensional (2D) magnetic semiconductors with room-temperature ferromagnetism is a significant challenge in materials science and is important for the development of next-generation spintronic devices. Herein, we demonstrate that a 2D semiconducting antiferromagnetic Cu-MOF can be endowed with intrinsic room-temperature ferromagnetic coupling using a ligand cleavage strategy to regulate the inner magnetic interaction within the Cu dimers. Using the element-selective X-ray magnetic circular dichroism (XMCD) technique, we provide unambiguous evidence for intrinsic ferromagnetism. Exhaustive structural characterizations confirm that the change of magnetic coupling is caused by the increased distance between Cu atoms within a Cu dimer. Theoretical calculations reveal that the ferromagnetic coupling is enhanced with the increased Cu-Cu distance, which depresses the hybridization between $3d$ orbitals of nearest Cu atoms. Our work provides an effective avenue to design and fabricate MOF-based semiconducting room-temperature ferromagnetic materials and promotes their practical applications in next-generation spintronic devices.

2D magnetic materials with new physical phenomena have been a central issue in the area of condensed matter physics and been extensively investigated in the past few years[1–5]. They offer opportunities for the research and development of new spintronics devices due to their confined carrier migration and heat diffusion in the 2D plane, which shows rich exotic properties, such as unique electron transport and valley-related properties[6–8]. For practical applications in next-generation data processing and storage devices, the 2D magnetic materials are required to be ferromagnetic and stable above room temperature[9,10]. Unfortunately, most of these materials cannot meet these requirements because the long-range magnetic orders are critically restrained by thermal fluctuations[11]. Although there have been attempts to engineer magnetic properties in 2D materials, such as atomic vacancies, interface or component engineering and chemical

doping[12–14], 2D materials with intrinsic ferromagnetism (FM) were only recently reported with insulating $CrI_3$, $Cr_2Ge_2Te_6$, and metallic $Fe_3GeTe_2$[15–17]. However, the weak coupling between free carriers (s or p electrons) and local spins (d electrons) causes low $T_C$, and extreme instability in the air, which has limited their development in scientific research and practical application[17,18]. Air-stable semiconducting 2D materials with room-temperature ferromagnetic order are rarer still. These inorganic magnets have several drawbacks, including the high-density, inflexibility, and limited chemical tunability, which cannot meet the demand of next-generation multifunctional microdevices.

Fortunately, 2D metal-organic frameworks (MOFs), as a new class of 2D materials, provide possibilities to solve these challenges. In MOFs, due to their synthetic programmability and structure diversity[19–22], various MOFs have been developed for chemical

[1]National Synchrotron Radiation Laboratory, University of Science and Technology of China, 230026 Hefei, Anhui, China. [2]Hefei National Laboratory for Physical Sciences at the Microscale, University of Science and Technology of China, 230026 Hefei, Anhui, China. ✉e-mail: hlduan@ustc.edu.cn; chaowng@ustc.edu.cn; ywsh2000@ustc.edu.cn

sensors[23], gas adsorption and separation[24], and electrocatalysis[25]. More interestingly, due to the possibility of high coercivity and $T_C$, low-density 2D MOFs have been widely studied for their magnetic properties, which are mainly attributed to the strong in-plane $\pi$-$d$ conjugation between metal ions and organic ligands[26–29]. For example, Perlepe et al. realized the ferromagnetism with $T_C$ up to 242 °C by post-synthetic reduction of coordination networks in 2D chromium pyrazine MOF[30]; Park et al. reported ferromagnetism with $T_C$ of 225 K in a mixed-valence chromium MOF material[31]. Despite these developments, the realization of 2D semiconducting MOF magnets with room-temperature ferromagnetic order still remains a challenge.

Among various MOFs, those constructed by the secondary building unit containing a Cu(II) dimer (Cu$_2$-SBUs) with semiconducting behavior are attractive candidates for achieving semiconducting room-temperature ferromagnetism. First, Cu$^{2+}$ provides the necessary source of magnetic moments for macroscopic magnetism. Second, the strong super-exchange interaction between two Cu(II) ions within a paddle-wheel also provides a bridge for the generation of macroscopic magnetism[32]. Finally, the extended high conjugated structure may promote stronger exchange coupling interactions between the adjacent spins due to the inherent structural property[33]. Moreover, the programmability of MOFs means that the orbital's energy and spin state can be elaborately adjusted by the synthesis and substitute of the appropriate organic linkers[28], which indicates that ligand cleavage and defect introduction strategies could be used to achieve semiconducting room-temperature magnetism in 2D Cu$_2$-SBU-constructed MOF materials.

In this work, we demonstrate that a 2D semiconducting anti-ferromagnetic Cu-MOF can be endowed with intrinsic room-temperature ferromagnetic coupling using a ligand cleavage strategy to regulate the inner magnetic interaction within the Cu dimers. We confirm the intrinsic ferromagnetism of the 2D Cu-MOF using XMCD measurements. Detailed atomic and electronic structure analysis confirms that the ferromagnetic coupling between Cu ions is modulated by the increased distance between them within a Cu dimer induced by the ligand cleavage. The ligand cleavage generates local strain for the Cu dimer and pushes the nearest Cu atoms away from each other. Magnetic measurements suggest that the FM coupling can persist up to room temperature. Furthermore, theoretical calculations simulate the change of electronic structure in the process of ligand cleavages and demonstrate that the room temperature ferromagnetism is due to the depressed hybridization between 3$d$ orbitals of nearest Cu atoms. Our work highlights that the semiconducting room-temperature ferromagnetism can be achieved in MOFs, and the MOF materials can be tailored for the applications in next-generation electronic devices.

## Results
### Sample preparations and structural characterizations
The Cu-ABDC MOF composed of Cu$_2$ paddle-wheel units was synthesized through a facile coordination reaction between 2-amino-1,4-benzenedicarboxylic acid (ABDC) and copper (II) nitrate (Cu(NO$_3$)$_2$)·3H$_2$O. Furthermore, the local strained Cu-ABDC MOFs (denoted as LS-Cu-ABDC) were obtained when 25% and 50% of ABDC ligands were replaced by benzoic acids as the linker molecules, which causes partial missing of organic ligand, thus results in the local swelling of the Cu-ABDC lattice due to the increased Cu-Cu distance within a Cu dimer (Fig. 1a and Supplementary Fig. 1). Experimental details are shown in Supplementary Methods section. The morphologies of the as-obtained samples are shown by transmission electron microscopy (TEM) and atomic force microscopy imaging (Fig. 1b, c and Supplementary Fig. 2). All the Cu-ABDC and LS-Cu-ABDC MOFs show 2D layered stacking nature with a lateral size of about 200 nm with the thickness of the nanosheets ~3.2 nm, revealing the ultrathin nature of the Cu-MOFs. Furthermore, the energy dispersive X-ray mapping

images (Fig. 1d and Supplementary Fig. 2) confirm that only Cu, O, C, and N elements are present in Cu-ABDC and LS-Cu-ABDC MOFs and no other impurities are detected, in accordance with inductively coupled plasma atomic emission spectroscopy (ICP-AES), and combustion elemental analysis (see details in Supplementary Table 1). X-ray diffraction (XRD) patterns in Fig.1e reveal the highly crystalline structures of Cu-ABDC and LS-Cu-ABDC. The Rietveld refinement of the XRD data (see details in Supplementary Fig. 3 and Supplementary Table 2) reveals all peaks of the MOF are indexed to the standard structure (CCDC-687690)[34], matching well with the reference diffraction patterns[34–36], indicating a reliable quality for our samples. In addition, the prominent peaks at ~17.0° are assigned to the (20-1) plane, suggesting that the stacking direction of the layers is perpendicular to this plane[34,37]. Furthermore, synchrotron powder X-ray diffraction analysis and solid-state cyclic voltammetry measurements were also performed (see details in Supplementary Figs. 3 and 4 and Supplementary Note 3), which excludes the formation of the Cu-related second phase. The high-resolution TEM (HRTEM) image in Fig. 1f shows a fringe lattice parameter of 1.48 nm for Cu-ABDC nanosheets, corresponding to the four lattice distance of (040) planes. After introducing benzoic acid to replace the ABDC linkers, the interlayer spacing is enlarged to 1.50 nm and 1.52 nm (Fig. 1g, h), respectively, corresponding to the local swelling ratios of 1.4 and 2.7% (denoted as 1.4%-LS-Cu-ABDC and 2.7%-LS-Cu-ABDC). Furthermore, the introduction of local strain could also be verified by the enlarged XRD patterns in Fig. 1e, where the diffraction peak of (040) plane at 25.1° shifts downwards to about 24.7° and 24.5°. Besides, Fourier-transformed infrared (FT-IR; Supplementary Fig. 5) and Raman spectroscopy (Supplementary Fig. 6) were also performed to confirm the successful introduction of local strain. It is noteworthy that, the cleaving strategy mainly alters the local coordinated environment, resulting in slight changes in the overall lattice structure of the MOF. Using a ligand cleavage strategy, we successfully introduce the local strain in thin layered Cu-ABDC MOF.

### Atomic and electronic structure analysis
In order to further investigate the changes of the atomic coordination environment after the ligand cleavage in Cu-ABDC MOF, X-ray absorption fine structure (XAFS) spectroscopy measurements were carried out. The Fourier transformed (FT) curves of the Cu K-edge extended X-ray absorption fine structure (EXAFS) $k^3\chi(k)$ functions for Cu-MOFs are summarized in Fig. 2a. All the curves display a prominent peak at about 1.5 Å, corresponding to the nearest coordination shell of the metal-oxygen bonds (Cu-O; Supplementary Fig. 7). The relatively weak peaks at about 2.0 Å and 2.7 Å could be assigned to the higher shell contribution of the Cu-Cu and Cu-O/C coordination, respectively[38], which can be further verified by the wavelet transform (WT) analysis (Fig. 2b and Supplementary Fig. 8). It is worth noting that the first peak for the Cu-O bond at about 1.5 Å slightly shifts to higher $R$ as the substitution of ligands is increased. Simultaneously, the second peak for the Cu-Cu coordination at about 2.00 Å moves to 2.02 Å and 2.04 Å, respectively. Furthermore, based on the actual crystal structure, there are two kinds of Cu-O coordination with different bond distances, including four radial sites (linked to organic ligand, denoted as Cu-O$_1$) and one axial site (linked to solvent molecular, denoted as Cu-O$_2$). Hence, the Cu K-edge EXAFS fitting analysis in $R$ range from 1.1 to 2.4 Å was employed by using Cu-O$_1$ (short), Cu-O$_2$ (long), and Cu-Cu backscattering paths to examine the variation of coordination environment of Cu ions (see details in Supplementary Figs. 9 and 10 and Supplementary Table 3). Obviously, the coordination number of Cu-O$_1$ decreases while Cu-O$_2$ increases with the increased substitution of ligands, which is mainly due to the dangling sites of Cu ions being spontaneously connected by nitrate or hydroxide ions to balance the charge[22,39]. The coordination number of Cu-Cu remains unchanged, but the distance between the Cu ions increases from 2.67 ± 0.01 Å in Cu-ABDC to 2.72 ± 0.01 Å and 2.75 ± 0.01 Å in 1.4%-LS-Cu-ABDC and

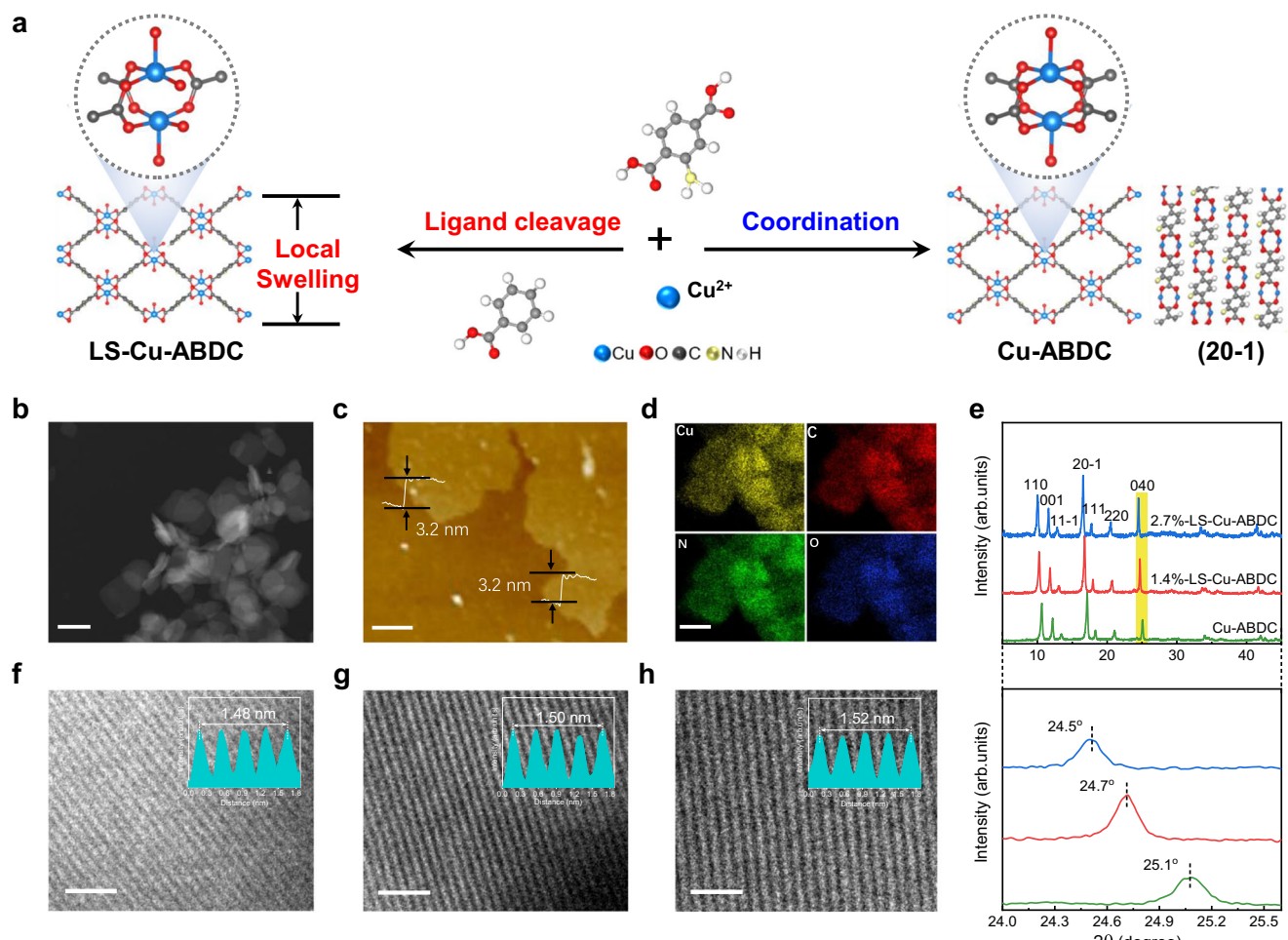

**Fig. 1 | Morphological and structural characterizations of Cu-MOFs. a** Schematic illustration of synthesis processes of the Cu-ABDC and LS-Cu-ABDC MOFs. The axial organic solve molecules are omitted for clarity in all structure models. **b** TEM image of the Cu-ABDC nanosheets. Scale bar, 200 nm. **c** Atomic force microscopy image of Cu-ABDC. Scale bar, 0.5 μm. **d** Corresponding elemental mapping images of Cu-ABDC. Scale bar, 100 nm. **e** Top panel: XRD patterns of all samples in 2θ ranges of 5–45°. Bottom panel: enlarged XRD patterns of (040) plane (marked as yellow shading). **f–h** HRTEM images of Cu-ABDC (**f**), 1.4%-LS-Cu-ABDC (**g**), and 2.7%-LS-Cu-ABDC (**h**), respectively. Scale bar, 2 nm.

2.7%-LS-Cu-ABDC, respectively (see details in Supplementary Table 3). These EXAFS results suggest that the introduction of benzoic acids leads to the increase of the distance of Cu-Cu within the dimers and thus the local expansion of interlayer spacing for the (040) plane in the LS-Cu-ABDC MOFs, consistent with the HRTEM and XRD results.

To further elucidate the influence of the ligand cleavage on the electronic structure of Cu ions, Cu L-edge, O K-edge X-ray absorption spectroscopy (XAS) and X-ray photoelectron spectroscopy (XPS) measurements were performed. The Cu L-edge XAS spectra show two characteristic peaks at about 930 eV ($L_3$) and 950 eV ($L_2$) (Fig. 2c), close to that of CuO (Supplementary Fig. 11), indicating the valence states of +2 for the Cu ions in Cu-ABDC and LS-Cu-ABDC[40]. The intensity of the $L_3$-edge peak gradually decreases with the increased substitution of ligands, suggesting the increased electron occupation of Cu 3*d* states in LS-Cu-ABDC MOFs, because the Cu L-edge XAS arises from the transition of Cu 2*p* electron to unoccupied 3*d* orbital, which is hybridized with O 2*p* orbital in Cu-MOFs. Furthermore, we also measured the O K-edge XAS spectra. In Fig. 2d, a sharp peak at about 532.3 eV is assigned to Cu-O bonding in Cu-MOFs (Supplementary Fig. 12)[41], and its intensity decreases with the ligand cleavages, suggesting the increased electron occupation of Cu 3*d* states[42]. In addition, Cu K-edge X-ray absorption near edge structure (XANES) spectra and first-derivative XAENS spectra also indicate that the electron occupation of Cu 3*d* states gradually increases with the ligand cleavages

(Supplementary Fig. 13). Moreover, the Cu $2p_{3/2}$ XPS spectra in Fig. 2e suggest that Cu ions mainly exist as +2 valence in Cu-ABDC[43], and the redshift of peak positions from 935.0 eV in Cu-ABDC to 934.8 and 934.6 eV in 1.4%- and 2.7%-LS-Cu-ABDC, respectively, indicate the slightly higher electron occupation of Cu 3*d* states in LS-Cu-ABDC, in agreement with the XAS results. Moreover, XPS survey spectra clearly show that there are only C, N, O, and Cu elements in all the Cu-MOF samples, and no other magnetic impurities are detected (Supplementary Fig. 14). Different from the activation process, after the substitution of ligands to regulate the distance between Cu atoms within a Cu dimer, the radial sites of Cu ions are directly connected by nitrate or hydroxide ions to balance the charge in the bottom-up polymerization process[44,45]. As verified by the EXAFS fitting results, due to the unchanged total coordination number of Cu ions and changed bond distance of Cu-Cu, the Cu ions still remain +2 valence and only electron transfer occurs after the ligand replacement.

## Characterization of semiconductor properties
The bulk electrical conductivity of Cu-ABDC was determined from a pelletized sample via a four-probe method (inset in Fig. 2f). As shown in Fig. 2f, the variable-temperature electrical conductivity (σ) of the bulk Cu-ABDC was positively correlated with temperature with a non-linear increase of electrical conductivity in the temperature range of 320 to 373 K, with electrical conductivity of about $2.3 \times 10^{-9}$ S m$^{-1}$ at

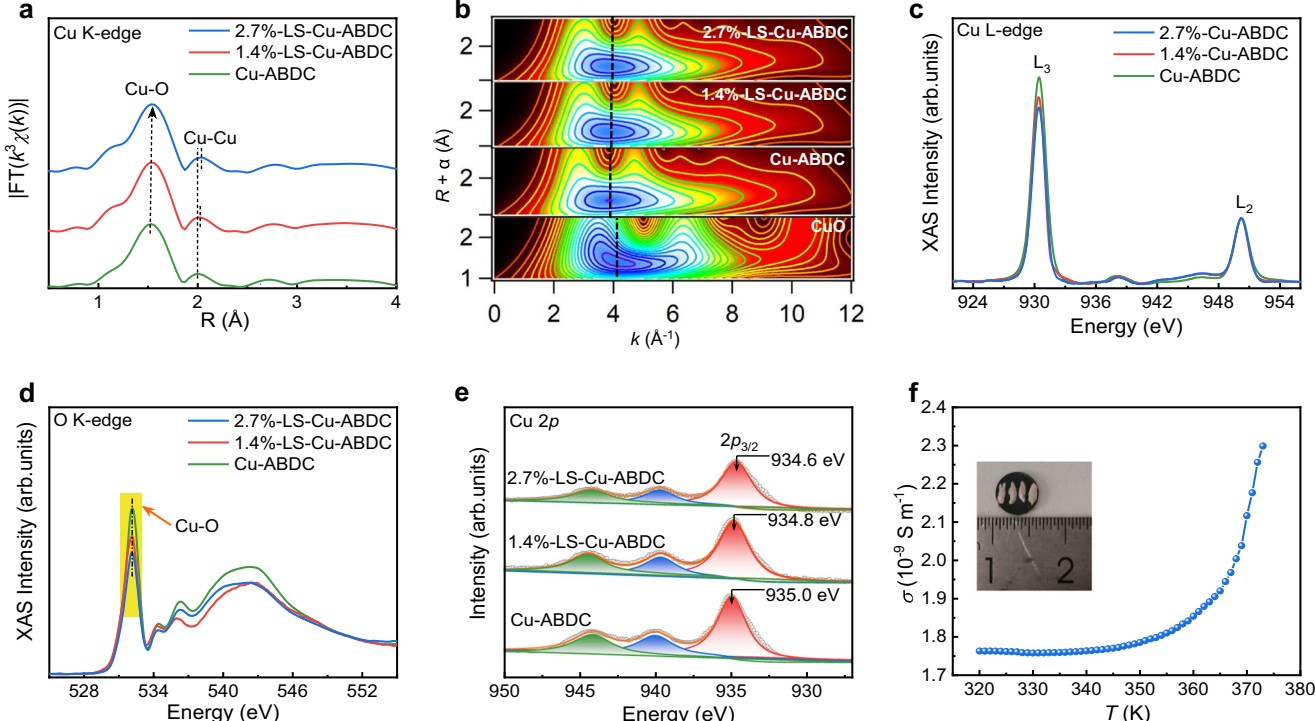

**Fig. 2 | Atomic and electronic structures of Cu-MOFs. a** Fourier transforms of the Cu K-edge EXAFS spectra for Cu-ABDC and LS-Cu-ABDC MOFs. **b** WT analysis of the first coordination shell for Cu-ABDC, LS-Cu-ABDC MOFs, and CuO. **c, d** Cu L-edge (**c**) and O K-edge (**d**) XAS spectra for Cu-ABDC and LS-Cu-ABDC MOFs (Cu-O interaction marked as yellow shading). **e** Cu 2p XPS spectra for Cu-ABDC and LS-Cu-ABDC MOFs. **f** Electrical conductivity (σ) of Cu-ABDC as a function of temperature (T).

373 K, which is typical for semiconducting materials[46–48]. It is worth noting that the observed pressed pellet conductivity values for low-dimensional materials are often artificially low[49,50]. These low values may arise from large anisotropy in the conducting pathway, which leads to unfavorable grain boundary interactions. As shown in Supplementary Fig. 15a, a linear relationship between the logarithm of conductivity (ln ($\sigma$)) and the inverse of temperature ($T^{-1}$) in the temperature range 365-373 K is displayed, as expected for thermally activated transport. Furthermore, by fitting to the Arrhenius equation, the activation energy ($E_a$) was found to be 0.3 eV[51,52]. The higher value of $E_a$ also suggests a lower conductivity. As shown in Supplementary Fig. 15b, another different conduction mechanism dominates at the lower temperature region from 355 to 365 K, which also shows a linear relationship between ln ($\sigma$) and $T^{-1/4}$. This behavior can be assigned to Mott variable-range hopping[53,54]. For all measurements, linear I-V curves were obtained (Supplementary Fig. 16). In addition, the ultraviolet-visible-near-infrared (UV-Vis-NIR) spectroscopy measurement, Mott-Schottky analysis, and photoconductivity measurements also provide further evidence for the semiconducting properties (see details in "Methods" section and Supplementary Figs. 17–19 and Supplementary Note 2).

**Magnetic properties of Cu-MOFs**

In order to explore the influence of the ligand cleavage on the magnetic coupling of Cu-MOFs, magnetic field (M-H) and temperature (M-T) dependent magnetization curves were measured using a superconducting quantum interference device (SQUID). Figure 3a and Supplementary Fig. 20a show the M-T and M-H curves of Cu-ABDC MOF at different temperatures. There is no divergence between zero-field cooled (ZFC) and field-cooled (FC) magnetization curves from 5 to 300 K under an applied field of 500 Oe, and negligible magnetic moment in M-H curves, except at low temperature, indicating antiferromagnetic order at high temperatures above 10 K

and possible ferromagnetic coupling at 5 K due to the spin canting effect or defect[38]. From the fitting results of $1/\chi$-T plot (magnetic susceptibility $\chi = M/H$, inset in Fig. 3a) at high-temperature region based on Curie-Weiss law $\chi = C/(T - \theta)$, a negative Weiss constant $\theta = -280$ K can be obtained, revealing the strong antiferromagnetic exchange interaction within a Cu dimer in Cu-ABDC, in agreement with previous reports[38,55]. In contrast, 1.4%-LS-Cu-ABDC and 2.7%-LS-Cu-ABDC show drastically different behavior, as indicated by the obvious divergence between FC and ZFC curves (Fig. 3b, c), and saturated magnetization and relative high remanent magnetization at the temperature up to 300 K (Supplementary Fig. 20b, c), suggesting the presence of ferromagnetism. These results indicate that the magnetic interaction between Cu atoms within a dimer can be tuned from AFM to FM by the increased distance between Cu ions within a dimer caused by the ligand cleavage. It is noteworthy that the splitting between FC and ZFC curves and saturated magnetization behavior with sizeable magnetic moment is observed up to 300 K for 1.4%- and 2.7%-LS-Cu-ABDC, suggesting that the ferromagnetic coupling can persist above room temperature. Such high Curie temperature has rarely been achieved by previous semiconducting MOFs[29,31,51]. M-H curves for LS-Cu-ABDC MOFs at 300 K are plotted in Fig. 3d to visually display the change of FM with ligand cleavages. It is noticeable that the magnetic moment is smaller than 1 $\mu_B$/Cu atom and the saturation magnetization and coercivity decreased when the lattice strain increases from 1.4% to 2.7%. This is due to the increased occupation of Cu 3d states. On one hand, the exchange interactions after the ligand replacement are usually weaker than that for four -(OCO)- bridges, and the strength of the interaction decreases with the number of intervening bonded atoms between the moment carriers[56,57]. On the other hand, the substitution of ligands is hard to precisely control and hence not uniform during sample preparation. It is possible that antiferromagnetic coupling remains in some Cu dimers whose ligands are not replaced, and their interactions result

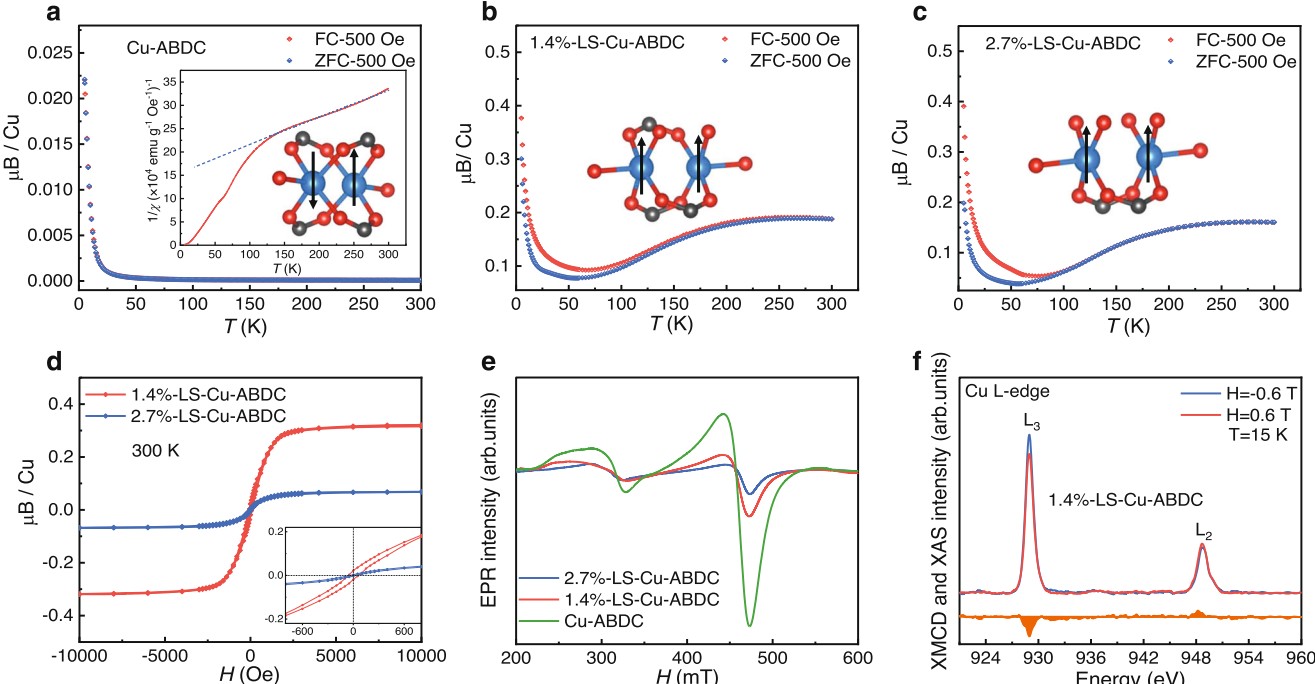

**Fig. 3 | Magnetic properties of the Cu-MOFs. a** *M-T* curves of Cu-ABDC with FC and ZFC process. Inset: 1/χ-*T* plot with Curie-Weiss fitting to the high-temperature region and the magnetic coupling within a dimer. **b, c** *M-T* curves with FC and ZFC process for 1.4%-LS-Cu-ABDC (**b**) and 2.7%-LS-Cu-ABDC (**c**). Inset: the magnetic coupling within a dimer of LS-Cu-MOFs. **d** Magnetic hysteresis loops at 300 K for 1.4%-LS-Cu-ABDC and 2.7%-LS-Cu-ABDC. **e** EPR signals for all the Cu-MOFs. **f** Cu L-edge XMCD measurements of 1.4%-LS-Cu-ABDC at 15 K.

in the "spin canting effect" and a decrease in the net magnetic moment such that the overall magnetic moment is smaller than 1 μB/Cu atom, in agreement with our theoretical calculations (Supplementary Table 4). This changed magnetic moment is further verified by the gradually reduced intensity of electron paramagnetic resonance (EPR) spectra (Fig. 3e), in which the peaks at about 330 mT and 470 mT are attributed to mononuclear and dinuclear $Cu^{2+}$ ions of Cu dimers[58,59]. Moreover, Cu L-edge X-ray magnetic circular dichroism (XMCD) measurements of 1.4%-LS-Cu-ABDC were carried out at 15 K to confirm the existence of ferromagnetic coupling between Cu ions (Fig. 3f). Significant and opposite XMCD signals (μ+ − μ−) were observed for the $L_3$ and $L_2$ edge, directly proving that the intrinsic FM originates from the spin polarization of Cu 3*d* electrons. In addition, we note that the magnetization of our samples increases from about 70 K to a local maximum at 270 K, in agreement with the competition between FM and AFM interactions, which has been studied as cooperative magnetism in complexes based on Cu dimer in the past[60,61]. For our MOF samples, the existence of AFM is possibly due to the nonuniform substitution of the ligands, and individual spins or small spin clusters are possibly formed due to the small size of our samples (~200 nm). It is difficult to precisely control and may be addressed by improving the sample preparation. However, we did not observe any frequency-dependent peaks in the AC magnetic susceptibility characterizations (see details in Supplementary Fig. 21), which are often attributed to short-range behavior, such as spin glass. Meanwhile, AC magnetic susceptibility signals related to long-range order were also missing in the measurement, probably due to the test temperature being well below the transition temperature $T_C$ or the response of our samples to AC magnetic susceptibility is relatively weak because of the weak magnetism. In combination with the element analysis, we can conclude that the magnetism in LS-Cu-ABDC MOFs is intrinsic and can be effectively tuned by the changed distance between Cu atoms within a Cu dimer caused by the ligand cleavage.

## Origin of the magnetism in Cu-MOFs

For an in-depth understanding of the influence of ligand cleavage induced on the magnetic coupling within a Cu dimer, we performed spin-polarized density functional theory (DFT) calculations using Quantum Espresso software package. The atomic models for DFT calculation are based on the crystal structure obtained from the above composition and structure analysis results (see calculation details in the section of "Methods" and Supplementary Note 1). The obtained exchange energy ($\Delta E = E_{FM} - E_{AFM}$, defined as the difference in energy between the FM and AFM spin configurations), magnetic moment and the distance between Cu ions within a dimer are respectively summarized in Supplementary Table 4. With the cleavage of ligand, the distance between the nearest Cu ions gradually increases, which will cause local swelling in Cu-ABDC, in accordance with the above experimental results. More interestingly, the Cu-ABDC prefers AFM ground state, while the LS-Cu-ABDC MOFs prefer FM ground state, that is to say, the exchange interaction varies from AFM to FM between Cu ions after the increased distance between Cu atoms within a Cu dimer caused by the ligand cleavage, consistent with the analysis of magnetism measurements. Simultaneously, the magnetic moment also reduces due to the increased electron occupation of Cu 3*d* states.

Furthermore, the electronic structures of Cu-ABDC at AFM state, 1.4%-LS-Cu-ABDC at FM state, and 2.7%-LS-Cu-ABDC with adjacent and diagonal ligands at FM state were calculated, respectively. The obtained total densities of states (TDOSs) are shown in Supplementary Fig. 22, and projected DOSs (PDOSs) of Cu atoms and spatial distribution of spin density are shown in Fig. 4a–h. For Cu-ABDC, the Fermi level locates inside the valence band, indicating a *p*-type semiconducting behavior, consistent with the semiconductor properties analysis (see details in "Methods" section and Fig. 2f and Supplementary Figs. 15–19)[62]. Obviously, the spin-up and spin-down PDOSs of $Cu_1$ and $Cu_2$ atoms within a dimer are completely symmetric to each other (Fig. 4a), revealing the AFM

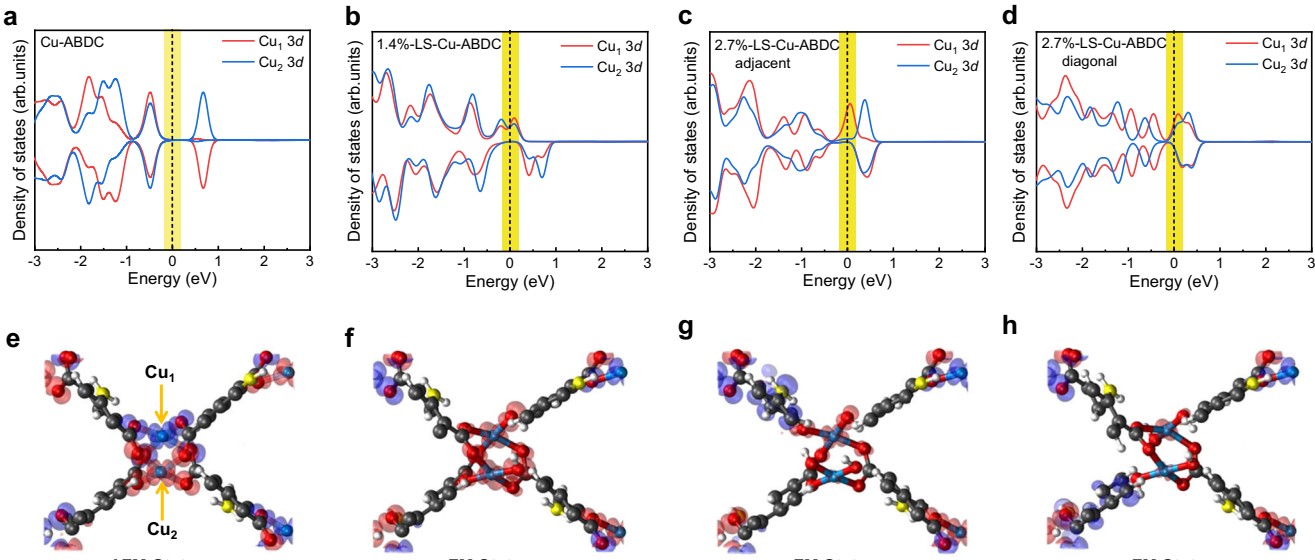

**Fig. 4 | Electronic structure calculations of Cu-MOFs. a–d** DFT calculated PDOSs of Cu atoms for Cu-ABDC (**a**), 1.4%-LS-Cu-ABDC (**b**), 2.7%-LS-Cu-ABDC with the adjacent (**c**), and diagonal (**d**) ligands, respectively. $Cu_1$ and $Cu_2$ represent the two Cu atoms within one dimer. **e–h** Corresponding spin density iso-surface distribution of a Cu dimer supercell for Cu-ABDC at AFM state (**e**), 1.4%-LS-Cu-ABDC at FM state (**f**), 2.7%-LS-Cu-ABDC with the adjacent (**g**), and diagonal (**h**) ligands at FM state, respectively. Red and blue iso-surfaces represent positive and negative spin densities, and the value is 0.002 spins per bohr[3], and the axial solvent molecular are omitted for clarity in all structure models. Highlight the position of the Fermi level with yellow shadings.

coupling between them[32]. The short bond-length (~2.6 Å) between nearest-neighboring Cu ions within the dimers results in the strong overlap of $d$ orbitals, thus more stable AFM interaction. More interestingly, after introducing ligand cleavage in Cu-ABDC, the spin-up and spin-down PDOSs of $Cu_1$ and $Cu_2$ atoms are no longer symmetric (Fig. 4b-d). The distance between Cu ions within a dimer increases gradually due to the ligand cleavage, leading to the spin polarization near the Fermi level, which favors FM coupling, mainly due to the decreased $d_{x^2-y^2}$ orbital hybridization between $Cu_1$ and $Cu_2$ atoms[63]. The gradually reduced magnetic moments are further validated by the decreased spatial distribution of spin densities (Fig. 4e–h). The TDOSs (see details in Supplementary Fig. 22) near the Fermi level display strong hybridization of the orbitals from Cu($d$), O($p$) and C($p$), suggesting a high degree of $\pi$-$d$ conjugation in the monolayer plane[31], in agreement with UV-Vis-NIR, and the theoretical calculations results (see details in Supplementary Figs. 17 and 23–24). Angle-dependent C K-edge XAS (see details in Supplementary Fig. 25) also confirms that delocalized $\pi$ electrons are present. Furthermore, computational results suggest long-range order in our sample, which may cause the changed neutron diffraction patterns after ligand cleavage (see details in Supplementary Figs. 26 and 27). Therefore, the indirect exchange interaction between the localized Cu spins may be mediated by the highly delocalized $\pi$ electrons[38,61,64], which could cause the long-range ferromagnetism.

## Discussion

In summary, taking the 2D semiconducting Cu-ABDC MOF as an example, we achieve intrinsic semiconducting room-temperature FM using a ligand cleavage strategy to regulate the Cu-Cu distance within a Cu dimer. Magnetic and XMCD measurements suggest that the FM coupling is intrinsic and can persist up to room temperature. As the Cu-Cu distance within Cu dimer increases with ligand cleavage, the hybridization of $d_{x^2-y^2}$ orbital is depressed and consequently the magnetic coupling between nearest atoms changes from AFM to FM. Our work provides a new semiconducting 2D material with room-temperature FM, and paves the way for the application of the 2D MOF materials in next-generation electronic devices.

## Methods

### Synthesis of Cu-ABDC and LS-Cu-ABDC MOFs

First, 64 mL N, N-dimethylformamide (DMF, Aladdin), 4 mL deionized water and 4 mL ethanol were mixed together in a 250 mL beaker. Then, 300 mg 2-amino-1, 4-benzenedicarboxylic acid (ABDC, Aladdin) and 200 mg $Cu(NO_3)_2 \cdot 3H_2O$ (Aladdin) were dissolved in the solution. After the precursors were dissolved, 1.6 mL triethylamine (TEA, Aladdin) was quickly added into the mixed solution and stirred for 5 min to obtain a uniform colloidal suspension. Afterwards, the colloidal suspension was continuously ultrasonicated for 8 h. Finally, the product Cu-ABDC nanosheets were obtained via centrifugation, washed with ethanol for 3–5 times, and dried at 80°C for 24 h. Besides, we used 60 mg benzoic acid (BA, Sinopharm Chemical Reagent) and 200 mg ABDC, and 90 mg BA and 130 mg ABDC to prepare 1.4%-LS-Cu-ABDC and 2.7%-LS-Cu-ABDC nanosheets, respectively. The subsequent processes are same to the preparation of Cu-ABDC nanosheets.

### Structure and property characterization

The transmission electron microscopy (TEM) measurements and corresponding energy-dispersive spectroscopy (EDS) mapping analyses were carried out on a JEM-2100F field emission electron microscope at an acceleration voltage of 80 kV. The high-resolution TEM (HRTEM) were performed on a JEOL JEM-2100F TEM/STEM at an acceleration voltage of 80 kV and under a low temperature about 80 K. The XRD and XPS patterns were detected on a Philips X'Pert Pro Super diffractometer with Cu $K\alpha$ line ($\lambda = 1.54178$ Å) and an ESCALAB MKII equipped with Mg $K\alpha$ source (hv = 1253.6 eV), respectively. UV-Vis-NIR were recorded on a Shimadzu DUV-3700 spectrophotometer. In the testing process of Raman spectra (Horiba HR Evolution), a 50× objective was used to focus incident 532 nm laser with the spot size of ~1 μm. The backscattered light was dispersed by a grating with 1200 grooves/mm. The magnetic impurities were excluded by inductively coupled plasma atomic emission spectrometry (ICP-AES, Jarrel Ash model 955). The electronic paramagnetic resonance (EPR) measurements were detected in a JSE-FA200 EPR spectrometer at X-band (~9 GHz) with a resolution of 2.35 μT at room temperatures. FTIR data were collected at the infrared beamline BL01B of the National Synchrotron Radiation Laboratory (NSRL, China). The Cu K-edge X-ray

absorption fine structure (XAFS) were measured at the 1W1B beamline of the Beijing Synchrotron Radiation Facility (BSRF) and BL14W beamline of the Shanghai Synchrotron Radiation Facility (SSRF), China. The synchrotron powder X-ray diffraction data were obtained at 1W1A Diffuse X-ray Scattering Station, Beijing Synchrotron Radiation Facility (BSRF-1W1A). And the Cu L-edge and O K-edge X-ray absorption (XAS) spectra were measured at the Beamlines MCD-A and MCD-B (Soochow Beamline for Energy Materials) at National Synchrotron Radiation Laboratory (NSRL, China). Neutron diffraction measurements at the energy-resolving neutron imaging spectrometer (ERNI) of the China Spallation Neutron Source (CSNS).

### Magnetic measurements

Magnetic measurements were performed using a SQUID (Superconducting Quantum Interference Design). Variable-temperature direct current (d.c.) magnetic susceptibility of the samples was measured using a quartz sample holder in zero-filed cooling and field cooling sequence with the applied magnetic field of 500 Oe with a temperature range of 5–300 K. In our magnetic characterizations, M–H curves were measured at the scan rate of the magnetic field 50 Oe/s in the range of 0–300 Oe, 200 Oe/s in the range of 400–3000 Oe and 2000 Oe/s in the range of 4000–10,000 Oe with the field up to 2 T, M-T curves were measured over a temperature range of 5–300 K at 500 Oe with the heating rate of 10–15 K/min. Variable-temperature alternating current (a.c.) magnetic susceptibility of the samples was measured at the temperature range of 50–360 K, and at an AC field of $H_{ac} = 2$ Oe with DC field $H_{dc} = 0$ under different frequencies ($f = 1, 10, 100, 500,$ and 1000 Hz).

### Synchrotron-based XAS measurements

The Cu K-edge X-ray absorption fine structure (XAFS) data were collected in the transmission mode and calibrated using Cu foil. During the measurement, samples were placed at room temperature. Both the Cu L-edge and O K-edge were measured by means of total electron yield (TEY), measuring the drain current as a function of the photon energy. Multiple scans were measured and averaged. To get a great signal-to-noise ratio for the absorption curves for the Cu L-edge X-ray magnetic circular dichroism (XMCD) experiments, we performed the following procedure: Firstly, the sample holder was cooled down to the lowest possible temperature and kept for 4 h to get stable temperature conditions, and XCMD measurements were carried out. Next, XMCD spectra were acquired eight times and averaged at the applied field positive and negative 0.6 T, respectively.

### Electronic conductivity measurement

The pressed pellets were prepared by adding about 20 mg samples (heated at 100 °C under argon overnight) in a 6 mm inner diameter split sleeve under the applied pressure of 20 MPa. No binder or conducting additive was added to the sample. The thickness of the pressed pellet was about 0.49 mm. Then, four-probe points were placed onto the top of the pressed pellets using conductive silver adhesive. All electrical transport measurements were based on the four-probe method using a probe station with a source meter Keithley 2614B. I−V curves were collected by scanning the current in the voltage range from +5 V to −5 V.

### DFT calculation details

The atomic structure relaxations and electronic structure calculations were carried out with the spin-polarized DFT calculations implemented in the Quantum Espresso software package[65]. The electron-ion interaction and electron exchange-correlation was described with projected augmented wave (PAW) method with a kinetic energy cutoff of 1020 eV and generalized gradient approximation (GGA) in the Perdew-Burke-Ernzerhof (PBE) parametrization. Then the DFT-D3 scheme was employed to process the long-range van der Waals interaction. The Brillouin zone was integrated with a $3 \times 2 \times 4$ k-grid. The

convergences for self-consistent field calculations and atomic structure optimizations were $2 \times 10^{-7}$ eV/atom and 0.05 eV/Å. Some calculation results were visualized with VMD software[66].

## Data availability

All data generated in this study are provided in the Supplementary Information/Source Data file. Source data are provided with this paper.

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

## Acknowledgements

This work was financially supported by the National Key Research and Development Program of China (2021YFA1600800, W.Y.), National

Natural Science Foundation of China (Grants No. 11975234, W.Y., 11775225, W.Y., U2032150, Z.Q., U1932211, W.Y., 12075243, Z.S., 12005227, H.T., 12275271, C.W., 12205305, W.Y., and 12105286, H.D.), the Users with Excellence Program of Hefei Science Center CAS (No. 2020HSC-UE002, W.Y., 2020HSC-CIP013, W.Y., 2021HSC-UE002, C.W., and 2021HSC-UE003, W.Y.), the Major science and technology project of Anhui Province (202103a05020025, W.Y.), the Key Program of Research and Development of Hefei Science Center, CAS (2021HSC-KPRD002, W.Y.), the Fundamental Research Funds for the Central Universities (WK 2310000103, C.W.), and partially carried out at the USTC Center for Micro and Nanoscale Research and Fabrication. The authors would like to thank Beijing Synchrotron Radiation Facility (BSRF), Shanghai Synchrotron Radiation Facility (SSRF), and Beamlines MCD-A and MCD-B (Soochow Beamline for Energy Materials) at NSRL for the synchrotron beamline. The authors also thank Yuran Niu and Lin Zhu at MAXPEEM beamline in MAX IV Lab for the discussion about the valence states, and Y. Soh in Paul Scherrer Institute for the discussion about the magnetic properties. A portion of this work is based on the data obtained at BSRF-1W1A. The authors gratefully acknowledge the cooperation of the beamline scientists at BSRF-1W1A beamline. Numerical computations were performed on Hefei advanced computing center.

## Author contributions

H.D., C.W. and W.Y. conceived the experiments and supervised the project; S.F., H.T., Y.W., Z.L., L.C., F.H., Y.C. and Z.S. performed the measurements; S.F., H.D., Z.Q., L.S., C.L. and X.L. performed the magnetism measurements; C.W. performed the DFT calculations; S.F., H.D., C.W. and W.Y. analyzed the results; S.F., H.D., C.W. and W.Y. wrote the paper with comments from all authors.

## Competing interests

The authors declare no competing interests.
