## [Peer Review File · Nature Communications]

Reviewers' Comments:

Reviewer #1:

Remarks to the Author:

The manuscript by Feng et al. utilizes ligand substitution to modify the magnetic properties of a Cu-ABDC MOF. In particular, a partial substitution of ABDC linkers with benzoic acid results in the reduced framework connectivity and expansion of the lattice volume. As the result of the altered coordination environment around the copper ions, LS-Cu-ABDC materials exhibit modified electronic and magnetic structures. The authors nicely demonstrate the successful synthesis of a new two-dimensional MOF, which may be of great interest to many researchers. Furthermore, this work serves to highlight the ongoing research interest to tune the magnetic properties of MOFs. However, additional experimental evidences are required to justify the conclusion of the manuscript. Also, the manuscript needs further efforts on data analysis and presentation. Upon resolving the issues identified below, this article may be worthy of further consideration for publication in Nature Communications.

1. In various points of the manuscript, the authors suggest semiconducting properties of their MOFs. While the authors attempt to address this claim with UV-vis and Mott-Schottky analyses and computational results, the analyses yield inconsistent description of the bandgap. For a complete description of the materials' electronic structures, I strongly believe temperature-dependent electronic conductivity data is a necessary piece of information. Conductivity values need to be clearly presented, and charge transport mechanism must be discussed.

2. The authors need to clearly identify the chemical composition of all samples. For example, 1,4%-LS-Cu-ABDC and 2.7%-LS-Cu-ABDC samples were obtained by using approximately 25 and 50 mol% of benzoic acid during the synthesis. However, attempt to quantify the actual ratio of ABDC and benzoic acid in their samples is missing. Furthermore, the ligand substitution most likely requires the presence of other anions such as hydroxide and/or nitrate and solvent molecules to fill up the coordination sphere of copper ions. All of these need to be quantified. Otherwise, it is unclear how magnetic data was processed without the accurate determination of the chemical formula.

3. The synthesis of all materials utilize water and trimethylamine. Given the previous reports utilizing this synthetic condition to obtain CuO particles, the authors need to investigate and rule out this possibility. This is especially important since small particles of CuO are known to exhibit ferromagnetism with open magnetic hysteresis at room-temperature. I strongly suggest including combustion elemental analysis and synchrotron powder x-ray diffraction data. In addition, authors may also consider other techniques such as solid-state cyclic voltammetry. Alternatively, authors should utilize anhydrous condition for the synthesis of the parent Cu-ABDC material and 1,4%-LS-Cu-ABDC and 2.7%-LS-Cu-ABDC samples, and investigate the reproducibility of the data.

4. The authors attribute the lattice expansion to the missing linker based on XRD, TEM, etc. While I agree with the interpretation, the authors need to clearly explain how the missing linker causes lattice expansion rather than contraction.

5. Fit parameters obtained from the Rietveld refinement need to be presented in the SI.

6. The authors need to provide a better explanation for the reduced saturation magnetization. Authors suggest this is due to the increased occupation of Cu 3d states. However, they disregard the possibility of copper(I) ions and suggest that all copper ions exist in the 2+ oxidation state. I find these claims contradictory. Ferromagnetically coupled copper(II) ions ($S = 1/2$) should have saturation magnetization value of $\sim 1\mu\text{B}$. Observed moments for all samples are significantly lower than expected. Furthermore, magnetization in 2.7%-LS-Cu-ABDC sample doesn't seem to have saturated at 10000 Oe and decreases with increasing field above ~ 5000 Oe, which needs a clear explanation.

7. The authors need to clearly state whether their samples exhibit long-ranged or short-ranged magnetic order. The missing linkers should reduce the magnetic correlation within the system, and benzene carboxylates do not provide strong superexchange interaction between the magnetic centers. Any magnetic order, if present as the authors claim, should be short-ranged. Under these assumptions, I'm skeptical about the use of RSG phenomenon to describe the magnetic properties of their samples as the term is applicable for systems that exhibit long-ranged magnetic order. As such, I strongly suggest finding an alternative interpretation of the magnetic properties. If the authors believe their samples exhibit long-ranged ferromagnetic order, proper references and additional experimental data must be provided. Some suggestions include out-of-phase AC

magnetic susceptibility data, neutron diffraction, computational results on a larger unit-cell, etc.

8. It is unclear which features in the TDOS plots at the Fermi level suggest strong hybridization of the Cu and ligand orbitals. Authors need to clearly state contributions from individual atoms and whether these contributing orbitals have π symmetry. Otherwise, I suggest deleting the sentence: "The TDOSs near the Fermi level...".

9. I'm skeptical about the statement: "Therefore, the indirect exchange interaction between the localized Cu spins...". The manuscript lacks the experimental evidences to support the delocalized π electrons. As such, I suggest removing the sentence.

10. The authors need to clearly describe experimental procedures for all of their material characterization, including Mott-Schottky analysis and structure determination in Fig S1.

Minor points

1. References need to be formatted properly.

2. For all data normalization performed, the authors should clearly state and explain the procedure.

Reviewer #2:

Remarks to the Author:

In the current manuscript, Yan and coworkers demonstrate that a 2D semiconducting antiferromagnetic Cu-MOF can be endowed with intrinsic room-temperature ferromagnetic coupling using a ligand cleavage strategy to regulate the inner magnetic interaction within the Cu dimers. The authors provide an unambiguous evidence for intrinsic ferromagnetism using the element-selective X-ray magnetic circular dichroism (XMCD) technique. Systematic characterizations as well as theoretical calculations confirm that the change of magnetic coupling is caused by the increased distance between Cu atoms within a Cu dimer. This work is interesting and might provide an effective avenue to design and fabricate MOF-based semiconducting room-temperature ferromagnetic materials. It is suitable for publication in Nat. Commun after addressing the following minor points:

1. In order to understand the magnetic properties clearly, more information should be given.

Which sample holder was used during the measurements in the SQUID? Moreover, please show the calculated details for the measured magnetic moment from emu to $\mu\text{B}/\text{Cu}$ for the M-H curves.

2. As shown in Fig. 3d, the saturated magnetization of the 1.4%-LS-Cu-ABDC is much higher than that of the 2.7%-LS-Cu-ABDC, please give more discussion.

Response to Reviewers and a Summary of Changes

Many thanks to the reviewers for having given us valuable comments on the manuscript of NCOMMS-22-46092 submitted to the **Nat. Commun.**:

Title: Intrinsic room-temperature ferromagnetism in a two-dimensional semiconducting metal-organic framework

Authors: Sihua Feng, Hengli Duan, Hao Tan, Fengchun Hu, Chaocheng Liu, Yao Wang, Zhi Li, Liang Cai, Yuyang Cao, Chao Wang, Zeming Qi, Li Song, Xuguang Liu, Zhihu Sun and Wensheng Yan

The entire comments from Reviewers

Reviewer 1:

The manuscript by Feng et al. utilizes ligand substitution to modify the magnetic properties of a Cu-ABDC MOF. In particular, a partial substitution of ABDC linkers with benzoic acid results in the reduced framework connectivity and expansion of the lattice volume. As the result of the altered coordination environment around the copper ions, LS-Cu-ABDC materials exhibit modified electronic and magnetic structures. The authors nicely demonstrate the successful synthesis of a new two-dimensional MOF, which may be of great interest to many researchers. Furthermore, this work serves to highlight the ongoing research interest to tune the magnetic properties of MOFs. However, additional experimental evidences are required to justify the conclusion of the manuscript. Also, the manuscript needs further efforts on data analysis and presentation. Upon resolving the issues identified below, this article may be worthy of further consideration for publication in Nature Communications.

1. In various points of the manuscript, the authors suggest semiconducting properties of their MOFs. While the authors attempt to address this claim with UV-vis and Mott-Schottky analyses and computational results, the analyses yield inconsistent description of the bandgap. For a complete description of the materials' electronic structures, I strongly believe temperature-dependent electronic conductivity data is a necessary piece of information. Conductivity values need to be clearly presented, and charge transport mechanism must be discussed.

2. The authors need to clearly identify the chemical composition of all samples. For

example, 1,4%-LS-Cu-ABDC and 2.7%-LS-Cu-ABDC samples were obtained by using approximately 25 and 50 mol% of benzoic acid during the synthesis. However, attempt to quantify the actual ratio of ABDC and benzoic acid in their samples is missing. Furthermore, the ligand substitution most likely requires the presence of other anions such as hydroxide and/or nitrate and solvent molecules to fill up the coordination sphere of copper ions. All of these need to be quantified. Otherwise, it is unclear how magnetic data was processed without the accurate determination of the chemical formula.

3. The synthesis of all materials utilize water and trimethylamine. Given the previous reports utilizing this synthetic condition to obtain CuO particles, the authors need to investigate and rule out this possibility. This is especially important since small particles of CuO are known to exhibit ferromagnetism with open magnetic hysteresis at room-temperature. I strongly suggest including combustion elemental analysis and synchrotron powder x-ray diffraction data. In addition, authors may also consider other techniques such as solid-state cyclic voltammetry. Alternatively, authors should utilize anhydrous condition for the synthesis of the parent Cu-ABDC material and 1,4%-LS-Cu-ABDC and 2.7%-LS-Cu-ABDC samples, and investigate the reproducibility of the data.

4. The authors attribute the lattice expansion to the missing linker based on XRD, TEM, etc. While I agree with the interpretation, the authors need to clearly explain how the missing linker causes lattice expansion rather than contraction.

5. Fit parameters obtained from the Rietveld refinement need to be presented in the SI.

6. The authors need to provide a better explanation for the reduced saturation magnetization. Authors suggest this is due to the increased occupation of Cu 3d states. However, they disregard the possibility of copper(I) ions and suggest that all copper ions exist in the 2+ oxidation state. I find these claims contradictory. Ferromagnetically coupled copper(II) ions ($S = 1/2$) should have saturation magnetization value of $\sim 1 \mu_B$. Observed moments for all samples are significantly lower than expected. Furthermore, magnetization in 2.7%-LS-Cu-ABDC sample doesn't seem to have saturated at 10000 Oe and decreases with increasing field above ~ 5000 Oe, which needs a clear explanation.

7. The authors need to clearly state whether their samples exhibit long-ranged or short-ranged magnetic order. The missing linkers should reduce the magnetic correlation within the system, and benzene carboxylates do not provide strong super-exchange interaction between the magnetic centers. Any magnetic order, if present as the authors

claim, should be short-ranged. Under these assumptions, I'm skeptical about the use of RSG phenomenon to describe the magnetic properties of their samples as the term is applicable for systems that exhibit long-ranged magnetic order. As such, I strongly suggest finding an alternative interpretation of the magnetic properties. If the authors believe their samples exhibit long-ranged ferromagnetic order, proper references and additional experimental data must be provided. Some suggestions include out-of-phase AC magnetic susceptibility data, neutron diffraction, computational results on a larger unit-cell, etc.

8. It is unclear which features in the TDOS plots at the Fermi level suggest strong hybridization of the Cu and ligand orbitals. Authors need to clearly state contributions from individual atoms and whether these contributing orbitals have π symmetry. Otherwise, I suggest deleting the sentence: "The TDOSs near the Fermi level..." .

9. I'm skeptical about the statement: "Therefore, the indirect exchange interaction between the localized Cu spins..." . The manuscript lacks the experimental evidences to support the delocalized π electrons. As such, I suggest removing the sentence.

10. The authors need to clearly describe experimental procedures for all of their material characterization, including Mott-Schottky analysis and structure determination in Fig S1.

Minor points

1. References need to be formatted properly.

2. For all data normalization performed, the authors should clearly state and explain the procedure.

Reviewer 2:

In the current manuscript, Yan and coworkers demonstrate that a 2D semiconducting antiferromagnetic Cu-MOF can be endowed with intrinsic room-temperature ferromagnetic coupling using a ligand cleavage strategy to regulate the inner magnetic interaction within the Cu dimers. The authors provide an unambiguous evidence for intrinsic ferromagnetism using the element-selective X-ray magnetic circular dichroism (XMCD) technique. Systematic characterizations as well as theoretical calculations confirm that the change of magnetic coupling is caused by the increased distance between Cu atoms within a Cu dimer. This work is interesting and might provide an effective avenue to design and fabricate MOF-based semiconducting room-temperature ferromagnetic materials. It is suitable for publication in Nat. Commun after addressing

the following minor points:

1. In order to understand the magnetic properties clearly, more information should be given. Which sample holder was used during the measurements in the SQUID? Moreover, please show the calculated details for the measured magnetic moment from emu to $\mu\text{B}/\text{Cu}$ for the M-H curves.

2. As shown in Fig. 3d, the saturated magnetization of the 1.4%-LS-Cu-ABDC is much higher than that of the 2.7%-LS-Cu-ABDC, please give more discussion.

Response to the Reviewers

Reply to Reviewer 1

Comments: The manuscript by Feng et al. utilizes ligand substitution to modify the magnetic properties of a Cu-ABDC MOF. In particular, a partial substitution of ABDC linkers with benzoic acid results in the reduced framework connectivity and expansion of the lattice volume. As the result of the altered coordination environment around the copper ions, LS-Cu-ABDC materials exhibit modified electronic and magnetic structures. The authors nicely demonstrate the successful synthesis of a new two-dimensional MOF, which may be of great interest to many researchers. Furthermore, this work serves to highlight the ongoing research interest to tune the magnetic properties of MOFs. However, additional experimental evidences are required to justify the conclusion of the manuscript. Also, the manuscript needs further efforts on data analysis and presentation. Upon resolving the issues identified below, this article may be worthy of further consideration for publication in Nature Communications.

Reply: Thank you very much for your positive comments. We have revised the manuscript according to your advice and hope it could be published in *Nat. Commun.*

1. Question: In various points of the manuscript, the authors suggest semiconducting properties of their MOFs. While the authors attempt to address this claim with UV-vis and Mott-Schottky analyses and computational results, the analyses yield inconsistent description of the bandgap. For a complete description of the materials' electronic structures, I strongly believe temperature-dependent electrical conductivity data is a necessary piece of information. Conductivity values need to be clearly presented, and charge transport mechanism must be discussed.

Reply: Thank you for your professional suggestions. In principle, the Mott-Schottky analysis is widely used to determine the carrier density and type, and the negative slopes of Mott-Schottky plots indicate that all of our samples are *p*-type semiconductors [Cheng *et al.*, *J. Phys. Chem. C* **2012**, *116*, 24060-24067; Liao *et al.*, *Nat. Nanotech.* **2014**, *9*, 69-73; Cai *et al.*, *J. Am. Chem. Soc.* **2015**, *137*, 2622-2627]. Furthermore, the bandgap of Cu-ABDC, 1.4%-LS-Cu-ABDC, and 2.7%-LS-Cu-ABDC obtained from the UV-Vis-NIR measurement are 1.20, 1.22, and 1.27 eV, respectively. However, based on the DFT theoretical calculation results (**Supplementary Figure 22**), the bandgap values are about 0.58, 0.71, and 0.76 (adjacent) and 0.84 eV (diagonal), respectively. Generally, the bandgap of the semiconductor is underestimated by DFT calculations based on the local density approximation (LDA) or generalized gradient approximation (GGA) due to the self-interaction error [Mak *et al.*, *Phys. Rev. Lett.* **2010**, *105*, 136805; Gu *et al.*, *Appl. Phys. Lett.* **2015**, *107*, 183301; Pathak *et al.*, *Nat. Commun.* **2019**, *10*, 1721]. Therefore, the semiconducting properties of our samples can be confirmed by the Mott-Schottky analysis, UV-Vis-NIR and DFT calculations. In order to further confirm the materials' electronic structures, temperature-dependent electrical conductivity and photoconductivity measurements were performed, and the electron transport mechanism was also discussed below:

(1) Temperature-dependent electrical conductivity measurement. The pressed pellets were prepared by adding about 20 mg samples (heated at 100°C under argon overnight) in a 6 mm inner diameter split sleeve under the applied pressure of 20 MPa. No binder or conducting additive was added to the sample. The thickness of the pressed pellet was about 0.49 mm. Then, four-probe points were placed onto the top of the pressed pellets using conductive silver adhesive (inset of **Figure 2f**). All electrical transport measurements were based on the four-probe method using a probe station with a source meter Keithley 2614B. *I-V* curves were collected by scanning the current in the voltage range from +5 V to -5 V. As shown in **Figure 2f**, the electrical conductivity (σ) of the bulk Cu-ABDC was positively correlated with temperature with a non-linear increase of electrical conductivity at the temperature range of 320-373 K, with electrical conductivity about 2.3×10^{-9} S m⁻¹ at 373K, which is typical for semiconducting materials [Dong *et al.*, *Nat. Commun.* **2018**, *9*, 2637; Park *et al.*, *J. Am. Chem. Soc.* **2018**, *140*, 14533-14537; Xie *et al.*, *J. Am. Chem. Soc.* **2018**, *140*, 7411-7414; Yang *et al.*, *Nat. Commun.* **2019**, *10*, 3260; Pham *et al.*, *J. Am. Chem. Soc.* **2022**, *144*, 10615-10621; Tian *et al.*, *Nature* **2023**, *615*, 56-61]. It is worth noting that the observed pressed pellet conductivity values for low-dimensional materials are often artificially

low [Jeon *et al.*, *J. Am. Chem. Soc.* **2016**, *138*, 6583; Sun *et al.*, *J. Am. Chem. Soc.* **2016**, *138*, 14772]. These low values may arise from large anisotropy in the conducting pathway, which leads to unfavorable grain boundary interactions. To determine the activation energy for charge transport, the experimental conductivity data were fit with the Arrhenius equation: $\sigma = \sigma_0 \exp(-E_a/kT)$, where σ_0 is the conductivity pre-factor, E_a is the activation energy and k is the Boltzmann constant (8.617×10^{-5} eV/K) [Huang *et al.*, *Nat. Commun.* **2015**, *6*, 7408; Dou *et al.*, *J. Am. Chem. Soc.* **2017**, *139*, 13608-13611; Dong *et al.*, *Nat. Commun.* **2018**, *9*, 2637]. As shown in **Supplementary Figure 15a**, it displays a linear relationship between the logarithm of conductivity ($\ln(\sigma)$) and the inverse of temperature ($1/T$) in the temperature range 365-373 K, as expected for thermally activated transport. Furthermore, the value of E_a was evaluated as 0.3 eV from the slope. The higher value of E_a also suggests a lower conductivity. As shown in **Supplementary Figure 15b**, another different conduction mechanism dominates at the lower temperature region from 355 to 365 K, which also shows a linear relationship between $\ln(\sigma)$ and $T^{-1/4}$. This behavior can be assigned to Mott variable-range hopping [Wang *et al.*, *Chem. Mater.* **27**, **2015**, 1285-1291; Dong *et al.*, *Nat. Commun.* **2018**, *9*, 2637; Liu *et al.*, *Adv. Mater.* **2022**, *34*, 2204140; Beloborodov *et al.*, *Rev. Mod. Phys.* **2007**, *79*, 469-518; Dou *et al.*, *J. Am. Chem. Soc.* **2017**, *139*, 13608-13611; Dong *et al.*, *Nat. Commun.* **2018**, *9*, 2637; Yang *et al.*, *Nat. Commun.* **2019**, *10*, 3260; Park *et al.*, *Nat. Chem.* **2021**, *13*, 594-598]. For all measurements, linear I - V curves were obtained in **Supplementary Figure 16**.

Figure 2f. Temperature-dependent electrical conductivity measurement for Cu-ABDC. Electrical conductivity (σ) as a function of temperature ranging from 320 K to 373 K.

Supplementary Figure 15. The Plot of $\ln \sigma$ versus the reciprocal of the temperature (T^{-1}) over the range from 320-373 K (a). The Plot of $\ln(\sigma)$ versus $T^{-1/4}$ over the temperature region 355-365 K (b). The dotted line is the experiment data, and the solid line is the fitting curve.

Supplementary Figure 16. I - V curves for Cu-ABDC collected at 320 K and 370 K.

(2) Photoconductivity measurements. To further verify the semiconducting property of our samples, the photoconductivity measurements were also performed (**Supplementary Figure 19**). The thickness of the pressed pellet was about 0.37 mm. And Au particles were deposited on the surface of the pressed pellet samples as electrodes via the sputtering method. The photo-electrical property of the as-fabricated Cu-ABDC was characterized by a probe station using a 4200A-SCS source meter (Keithley) with Clarius software under 2-probe voltage linear scan mode. The incident light source was LED with wavelength of 1000 nm, and the light intensity was calibrated using an optical power meter (Newport Model No. 2936 R). The photoconductivity characterization was carried out in ambient condition by applying a sweeping bias of 10 V at room temperature. As shown in **Supplementary Figure 19**, the current values increase under 1000 nm illuminations, further indicating the

semiconducting character of Cu-ABDC [Castaldelli et al., *Nat. Commun.* **2017**, *8*, 2139; Liu et al., *Angew. Chem. Int. Ed.* **2019**, *58*, 9590-9595; Arora et al., *Adv. Mater.* **2020**, *32*, 1907063; Liu et al., *Adv. Mater.* **2022**, *34*, 2204140].

Supplementary Figure 19. The photo-response of Cu-ABDC under 1000 nm illumination.

Accordingly, the temperature-dependent electrical conductivity measurement for Cu-ABDC has been added in page 8 of the main text: “**Characterization of semiconductor properties.** The bulk electrical conductivity of Cu-ABDC was determined from a pelletized sample via the four-probe method (inset in Fig. 2f). As shown in Fig. 2f, the variable-temperature electrical conductivity (σ) of the bulk Cu-ABDC was positively correlated with temperature with a non-linear increase of electrical conductivity at the temperature range of 320-373 K, with electrical conductivity of about $2.3 \times 10^{-9} \text{ S m}^{-1}$ at 373K, which is typical for semiconducting materials. It is worth noting that the observed pressed pellet conductivity values for low-dimensional materials are often artificially low. These low values may arise from large anisotropy in the conducting pathway, which leads to unfavorable grain boundary interactions. As shown in Supplementary Fig. 15a, a linear relationship between the logarithm of conductivity ($\ln(\sigma)$) and the inverse of temperature (T^{-1}) in the temperature range 365-373 K is displayed, as expected for thermally activated transport. Furthermore, by fitting to the Arrhenius equation, the activation energy (E_a) was found to be 0.3 eV. The higher value of E_a also suggests a lower conductivity. As shown in the inset of Supplementary Fig. 15, another different conduction mechanism dominates at the lower temperature region from 355 to 365 K, which also shows a linear relationship between $\ln(\sigma)$ and $T^{-1/4}$. This behavior can be assigned to Mott variable-range hopping. For all measurements, linear I-V curves were obtained in

Supplementary Fig. 16. In addition, the ultraviolet-visible-near-infrared (UV-Vis-NIR) spectroscopy measurement, Mott-Schottky analysis and photoconductivity measurements also provide further evidence for the semiconducting properties (see details in Methods and Supplementary Figs. 17-19).”

2. Question: The authors need to clearly identify the chemical composition of all samples. For example, 1,4%-LS-Cu-ABDC and 2.7%-LS-Cu-ABDC samples were obtained by using approximately 25 and 50 mol% of benzoic acid during the synthesis. However, attempt to quantify the actual ratio of ABDC and benzoic acid in their samples is missing. Furthermore, the ligand substitution most likely requires the presence of other anions such as hydroxide and/or nitrate and solvent molecules to fill up the coordination sphere of copper ions. All of these need to be quantified. Otherwise, it is unclear how magnetic data was processed without the accurate determination of the chemical formula.

Reply: (1) According to the previous reports, the presence of the distinct sequences of functionalities along the MOF backbone will cause an abundant pore environment, and not change the crystal structure of the MOF [Deng *et al.*, *Science* **2010**, 327,846-850; Matthew *et al.*, *Chem. Commun.* **2001**, 2532-2533]. Therefore, the structure of Cu-ABDC is consistent with that of well-known monoclinic Cu(tpa)·DMF, which has been confirmed by our XRD results [Carson *et al.*, *Eur. J. Inorg. Chem.* **2009**, 16, 2338-2343; Tania *et al.*, *Nat. Mater.* **2015**, 14, 48-55; Zhan *et al.*, *Adv. Funct. Mater.* **2019**, 29, 1806720]. The empirical chemical of Cu(ABDC)·DMF for our Cu-ABDC sample can be obtained [Kitagawa *et al.*, *Angew. Chem. Int. Ed.* **2004**, 43, 2334-2375; Carson *et al.*, *Eur. J. Inorg. Chem.* **2009**, 16, 2338-2343; Arslan *et al.*, *J. Am. Chem. Soc.* **2011**, 133, 8258-8168; Carson, *et al.*, *Eur. J. Inorg. Chem.* **2014**, 2140-2145; Tania *et al.*, *Nat. Mater.* **2015**, 14, 48-55; Zhan *et al.*, *Adv. Funct. Mater.* **2019**, 29, 1806720; Jose *et al.*, *J. Phys. Chem. C* **2021**, 125, 22837-22847]. In order to check the validity of the Cu(ABDC)·DMF formula, we performed elemental analysis by a combination of inductively coupled plasma atomic emission spectrometry (ICP-AES) and C, N and H combustion methods, the weight percent is presented in the table below:

Table R1. Elemental analysis by a combination of ICP-AES and C, N and H combustion method for Cu-ABDC.

Cu-ABDC	N (%)	C (%)	H (%)	Cu (%)	O (%)
Experimental	8.78	41.94	3.95	19.71	25.62
Calculated	8.86	41.77	3.80	20.25	25.32

Taking into account of the formula $\text{CuC}_{11}\text{N}_2\text{O}_5\text{H}_{12}$, we can obtain the following elemental ratios according to the experiment results: $\text{C}/\text{H}\approx 11/12.4$, $\text{C}/\text{N}\approx 11/2$, $\text{C}/\text{Cu}\approx 11/1$, $\text{C}/\text{O}\approx 11/5$, which is basically consistent with the theoretical value. Therefore, the element analysis finally validates a chemical formula as $\text{Cu}(\text{ABDC})\cdot\text{DMF}$ for the layered semiconducting MOF that we have synthesized.

(2) After the substitution of ligands to introduce lattice strain, the axial sites of Cu ions are spontaneously connected by hydroxide ions to balance the charge in the bottom-up polymerization process [Chen *et al.*, *J. Am. Chem. Soc.* **2000**, *122*, 11559-11560; Marx *et al.*, *J. Catal.* **2011**, *281*, 76-87; Fang *et al.*, *J. Am. Chem. Soc.* **2014**, *136*, 9627-9636; Zhan *et al.*, *Adv. Funct. Mater.* **2019**, *29*, 1806720; Huang *et al.*, *Nat. Commun.* **2019**, *10*, 2779; Zhou *et al.*, *Adv. Mater.* **2021**, *33*, 2104341]. Thus, if one organic linker is substituted by benzoic acid, there will be one -OH attached to the Cu ion, and the content of -NH₂ will decrease because only the organic ligand ABDC has -NH₂ group in all the used components (ABDC, DMF, BA, triethylamine, water, ethyl alcohol). Thus, we can determine the actual ratio of ABDC and benzoic acid via the content of -NH₂ group. According to the results of the Fourier-transformed infrared (FT-IR) spectra (**Supplementary Figure 5**), we normalized the curves of all the samples by the vibration intensity of C=C, which is not changed during the benzoic acid substitution process. The peak at about 3482 cm⁻¹ is clearly observed, which is ascribed to the asymmetrical stretching vibration adsorption of the amine (-NH₂) groups [Mihaylov *et al.*, *J. Phys. Chem. C* **2016**, *120*, 23584-23595; Xie *et al.*, *Adv. Mater.* **2023**, 2212118]. Obviously, the vibration intensities of -NH₂ for 1.4%- and 2.7%-LS-Cu-ABDC gradually decrease with the increase of ligand cleavages by about 25% and 42%, respectively, which is close to the actual ratio of benzoic acid. Therefore, the chemical formula for 1.4%-LS-Cu-ABDC and 2.7%-LS-Cu-ABDC are $\text{Cu}(\text{ABDC})_{0.75}\cdot(\text{BA})_{0.25}\cdot\text{DMF}\cdot\text{OH}_{0.25}$ and $\text{Cu}(\text{ABDC})_{0.58}\cdot(\text{BA})_{0.42}\cdot\text{DMF}\cdot\text{OH}_{0.42}$, respectively.

Supplementary Figure 5. FT-IR analysis. The curves of all the samples are normalized by the vibration intensity of C=C for the same amount of the organic ligand. The peak at about 3482 cm^{-1} is clearly observed, which is ascribed to the asymmetrical stretching vibration adsorption of the amine ($-\text{NH}_2$) groups. Obviously, the vibration intensities of $-\text{NH}_2$ for 1.4%- and 2.7%-LS-Cu-ABDC gradually decrease with the increase of ligand cleavages by about 25% and 42%, respectively, which is close to the actual ratio of benzoic acid.

(3) We also performed elemental analysis for LS-Cu-ABDC MOFs by a combination of ICP-AES and C, N and H combustion method, the results of weight percent are presented in the table below:

Supplementary Table 1. Elemental analysis by a combination of inductively coupled plasma atomic emission spectrometry (ICP-AES) and C, N and H combustion method for Cu-ABDC and LS-Cu-ABDC MOFs.

Cu-ABDC	N (%)	C (%)	H (%)	Cu (%)	O (%)
Experimental	8.78	41.94	3.95	19.71	25.62
Calculated	8.86	41.77	3.80	20.25	25.32
1.4%-LS-Cu-ABDC	N (%)	C (%)	H (%)	Cu (%)	O (%)
Experimental	8.09	40.48	3.94	19.87	27.62
Calculated	7.90	41.61	3.95	20.65	25.81
2.7%-LS-Cu-ABDC	N (%)	C (%)	H (%)	Cu (%)	O (%)
Experimental	8.76	41.63	4.00	19.91	25.70
Calculated	7.50	43.04	2.85	21.69	24.84

Taking into account of the corresponding formula for 1.4%- and 2.7%-LS-Cu-ABDC MOFs according to the results of FT-IR analysis, we can obtain the following elemental ratios according to the experiment results: C/H \approx 10.75/12.59, C/N \approx 10.75/1.84, C/O \approx 10.75/5.1, C/Cu \approx 10.75/1 for 1.4%-LS-Cu-ABDC, and C/H \approx 10.58/12.20, C/N \approx 10.58/1.90, C/O \approx 10.58/4.9, C/Cu \approx 10.58/1 for 2.7%-LS-Cu-ABDC, which is basically consistent with the following chemical formula: Cu(ABDC)_{0.75}·(BA)_{0.25}·DMF·OH_{0.25} and Cu(ABDC)_{0.58}·(BA)_{0.42}·DMF·OH_{0.42}. Therefore, all the above results indicate that the chemical formula for Cu-ABDC, 1.4%- and 2.7%-LS-Cu-ABDC is Cu(ABDC)·DMF, Cu(ABDC)_{0.75}·(BA)_{0.25}·DMF·OH_{0.25} and Cu(ABDC)_{0.58}·(BA)_{0.42}·DMF·OH_{0.42}, respectively.

Accordingly, the original description in Supplementary Figure 4 in page 5 of the Supplementary Information, “*The curves of all the samples are normalized by the vibration intensity of C=C for the same amount of the organic ligand. For all the Cu-MOFs, the peak at about 3482 cm⁻¹ is clearly observed and ascribed to the asymmetrical stretching vibration adsorption of the amine groups. In the lower frequency region, the peaks at about 1590, 1507 and 1259 cm⁻¹ correspond to the N-H, C-H and C-N stretching bonds, respectively. The peaks at about 1101, 1500 and 1668 cm⁻¹ are ascribed to the vibrations of C-O, C=C stretching of the benzene ring and C=O of solvent molecule. In addition, two peaks at ~1384 and 1619 cm⁻¹ are assigned to the symmetric and asymmetric vibrations of the -COO groups, confirming the existence of ligand in Cu-MOFs skeleton. Besides, a characteristic peak at about 580 cm⁻¹ can be assigned to Cu-O stretching, suggesting that organic ligands are efficiently coordinated to Cu atoms to form Cu-MOFs. Moreover, the vibration intensities of -COO gradually decrease with the increase of ligand cleavages after the normalization with the signals of C=C, suggesting the missing of organic linkers.*” **has been changed to** Supplementary Figure 5 in page 8, “*The curves of all the samples are normalized by the vibration intensity of C=C for the same amount of the organic ligand. For all the Cu-MOFs, the peak at about 3482 cm⁻¹ is clearly observed which is ascribed to the asymmetrical stretching vibration adsorption of the amine groups. In the lower frequency region, the peaks at about 1590, 1507 and 1259 cm⁻¹ correspond to the N-H, C-H and C-N stretching bonds, respectively. The peaks at about 1101, 1500 and 1668 cm⁻¹ are ascribed to the vibrations of C-O, C=C stretching of the benzene ring and C=O of solvent molecule. In addition, two peaks at ~1384 and 1619 cm⁻¹ are assigned to the symmetric and asymmetric vibrations of the -COO groups, confirming the existence of*”

ligand in Cu-MOFs skeleton. Besides, a characteristic peak at about 580 cm^{-1} can be assigned to Cu-O stretching, suggesting that organic ligands are efficiently coordinated to Cu atoms to form Cu-MOFs. Moreover, the vibration intensities of -COO gradually decrease with the increase of ligand cleavages after the normalization with the signals of C=C, suggesting the missing of organic linkers. Obviously, the vibration intensities of -NH₂ for 1.4%- and 2.7%-LS-Cu-ABDC gradually decrease with the increase of ligand cleavages by about 25% and 42%, respectively, which is close to the actual ratio of benzoic acid. Therefore, the chemical formula for 1.4%-LS-Cu-ABDC and 2.7%-LS-Cu-ABDC samples can be estimated to be $\text{Cu}(\text{ABDC})_{0.75}\cdot(\text{BA})_{0.25}\cdot\text{DMF}\cdot\text{OH}_{0.25}$ and $\text{Cu}(\text{ABDC})_{0.58}\cdot(\text{BA})_{0.42}\cdot\text{DMF}\cdot\text{OH}_{0.42}$, respectively.” and the original description in Supplementary Table 1 in page 20 of the Supplementary Information, “There are no other magnetic impurities in our samples, excepting Cu²⁺ ions, and the concentration of the Cu ions remain nearly unchanged in our MOF. The initial sample concentration was 10 mg / L. Based on the chemical formula $\text{Cu}_2(\text{ABDC})\cdot 2\text{DMF}$, the theoretical weight percent of Cu ions is 28%, which is close to our experiment results.” **has been changed** to Supplementary Table 1 in page 30 of the Supplementary Information. “In order to check the validity of the $\text{Cu}(\text{ABDC})\cdot\text{DMF}$ formula, we performed elemental analysis by a combination of inductively coupled plasma atomic emission spectrometry (ICP-AES) and C, N and H combustion methods, the weight percent is presented in the table. Taking into account of the formula $\text{CuC}_{11}\text{N}_2\text{O}_5\text{H}_{12}$, we can obtain the following elemental ratios according to the experiment results: $\text{C}/\text{H}\approx 11/12.4$, $\text{C}/\text{N}\approx 11/2$, $\text{C}/\text{Cu}\approx 11/1$, $\text{C}/\text{O}\approx 11/5$, which is basically consistent with the results of chemical formula. Therefore, the element analysis finally validates a chemical formula as $\text{Cu}(\text{ABDC})\cdot\text{DMF}$ for the layered semiconducting MOF that we have synthesized. We also performed elemental analysis for LS-Cu-ABDC MOFs. Taking into account of the corresponding formula for 1.4%- and 2.7%-LS-Cu-ABDC MOFs according to the results of FT-IR analysis, we can obtain the following elemental ratios according to the experiment results: $\text{C}/\text{H}\approx 10.75/12.59$, $\text{C}/\text{N}\approx 10.75/1.84$, $\text{C}/\text{O}\approx 10.75/5.1$, $\text{C}/\text{Cu}\approx 10.75/1$ for 1.4%-LS-Cu-ABDC, and $\text{C}/\text{H}\approx 10.58/12.20$, $\text{C}/\text{N}\approx 10.58/1.90$, $\text{C}/\text{O}\approx 10.58/4.9$, $\text{C}/\text{Cu}\approx 10.58/1$ for 2.7%-LS-Cu-ABDC, which is basically consistent with the following chemical formula: $\text{Cu}(\text{ABDC})_{0.75}\cdot(\text{BA})_{0.25}\cdot\text{DMF}\cdot\text{OH}_{0.25}$ and $\text{Cu}(\text{ABDC})_{0.58}\cdot(\text{BA})_{0.42}\cdot\text{DMF}\cdot\text{OH}_{0.42}$. Therefore, all above results indicate that the chemical formula for Cu-ABDC, 1.4%- and 2.7%-LS-Cu-ABDC can be estimated to be $\text{Cu}(\text{ABDC})\cdot\text{DMF}$, $\text{Cu}(\text{ABDC})_{0.75}\cdot(\text{BA})_{0.25}\cdot\text{DMF}\cdot\text{OH}_{0.25}$ and $\text{Cu}(\text{ABDC})_{0.58}\cdot(\text{BA})_{0.42}\cdot\text{DMF}\cdot\text{OH}_{0.42}$, respectively. Besides, the atomic ratio of C/Cu is

in agreement with the theoretical results, indicating no excess Cu. Therefore, there is also no CuO in our samples.”

3. Question: The synthesis of all materials utilize water and trimethylamine. Given the previous reports utilizing this synthetic condition to obtain CuO particles, the authors need to investigate and rule out this possibility. This is especially important since small particles of CuO are known to exhibit ferromagnetism with open magnetic hysteresis at room temperature. I strongly suggest including combustion elemental analysis and synchrotron powder X-ray diffraction data. In addition, authors may also consider other techniques such as solid-state cyclic voltammetry. Alternatively, authors should utilize anhydrous condition for the synthesis of the parent Cu-ABDC material and 1.4%-LS-Cu-ABDC and 2.7%-LS-Cu-ABDC samples, and investigate the reproducibility of the data.

Reply: Thank you for your professional suggestions. To exclude the existence of the CuO particles in our MOF samples, we carried out the combustion elemental analysis, synchrotron radiation powder X-ray diffraction and solid-state cyclic voltammetry measurements. Besides, we also utilize anhydrous condition to synthesize the Cu-ABDC, 1.4%-LS-Cu-ABDC and 2.7%-LS-Cu-ABDC samples to probe their electronic structures and magnetic properties. All the results are summarized below:

(1) Combustion elemental analysis. In order to check the validity of the Cu(ABDC)·DMF formula and confirm the absence of CuO, we performed elemental analysis by a combination of ICP-AES and C, N and H combustion method, the weight percent results are presented in **Supplementary Table 1** above. From these results, and taking into account of the corresponding formula for Cu-ABDC and LS-Cu-ABDC MOFs, we can obtain the following elemental ratios: C/Cu = 11/1, 10.75/1 and 10.58/1, respectively, in agreement with the theoretical value, indicating no excess Cu. Therefore, there is no CuO in our samples.

(2) Synchrotron radiation powder X-ray diffraction analysis. In order to further confirm the absence of CuO particles in our samples, synchrotron radiation powder X-ray diffraction analysis was carried out at 1W1A Diffuse X-ray Scattering Station, Beijing Synchrotron Radiation Facility (BSRF-1W1A). As shown in **Supplementary Figure 3b**, compared to the characteristic peak at about 35.5° and 38.7° for CuO, there are no obvious peaks at the same position for 1.4%-LS-Cu-ABDC sample, confirming the absence of CuO and hence intrinsic ferromagnetism in our samples. Besides, we also synthesized the 1.4%-LS-Cu-ABDC sample using anhydrous condition, which

shows the same characteristic diffraction peaks as our 1.4%-LS-Cu-ABDC sample (**Supplementary Figure 3b**), indicating that the presence of water in the solvent will not influence the crystal structure of Cu-ABDC, in agreement with previous reports [Chen *et al.*, *Angew. Chem. Int. Ed.* **2005**, *44*, 4745-4749; Rodenas *et al.*, *Nat. Mater.* **2015**, *14*, 48-55; Dissegna *et al.*, *Adv. Mater.* **2018**, *30*, 1704501; Zhou *et al.*, *Adv. Mater.* **2021**, *33*, 2104341].

Supplementary Figure 3b. Synchrotron radiation powder X-ray diffraction patterns of 1.4%-LS-Cu-ABDC and the reference samples.

(3) Solid-state cyclic voltammetry (CV) measurement. The solid-state CV measurements were performed on a CHI760D electrochemical workstation using a typical three-electrode system. The Cu-ABDC nanosheets on glassy carbon electrodes acted as the working electrode with a platinum mesh as the counter electrode and an Ag/AgCl reference electrode. Typically, 4 mg of sample and 30 μ L Nafion solution (5 wt%, Sigma Aldrich) were dispersed in 1 mL ethanol solution to form a homogeneous ink assisted by ultrasonic method. The CV curves (**Supplementary Figure 4**) were measured at 0.1 V/s scan rate in 0.1 M TBAPF₆/DMF (TBAPF₆, Tetrabutylammonium Hexafluorophosphate) and KCl solutions. As shown in **Supplementary Figure 4**, the peaks at about -0.37 and -0.47 V for Cu-ABDC MOFs, and peaks at about -0.21 and -0.47 V for CuO were observed in 0.1 M TBAPF₆/DMF solution, indicating the successful synthesis of Cu-ABDC MOFs and the absence of CuO. In 0.1 M KCl solution, two pairs of redox peaks were observed for Cu-ABDC MOFs, and only one pair of redox peaks at about 0.12 and -0.13 V for CuO. These different peak positions between CuO and Cu-MOFs indicate the absence of CuO nanoparticles in our samples [Souto *et al.*, *J. Am. Chem. Soc.* **2018**, *140*, 10562-10569; Zampardi *et al.*, *Small* **2018**, *14*, 1801765; Souto *et al.*, *Chem. Sci.* **2018**, *9*, 2413; Pathak *et al.*, *Nat. Commun.* **2019**, *10*, 1721].

Supplementary Figure 4. Solid-state CV curves of Cu-ABDC MOFs and CuO using 0.1 M TBAPF₆/DMF and KCl solutions as electrolytes at 0.1 V/s scan rate.

(4) Anhydrous condition experiments. We also utilize anhydrous condition to synthesize the Cu-ABDC, 1.4%-LS-Cu-ABDC and 2.7%-LS-Cu-ABDC samples, and give the results of valence state and magnetic properties (**Figure R1**). From the results of X-ray absorption spectroscopy (XAS) at Cu *L*-edge and O *K*-edge, we can conclude that the valence state and coordination environment of Cu ions is unaffected by the anhydrous condition. Furthermore, anhydrous 1.4%-LS-Cu-ABDC sample also exhibits ferromagnetic behavior at room temperature.

Figure R1. Electronic structure and magnetic properties for anhydrous 1.4%-LS-Cu-ABDC sample.

Accordingly, the combustion elemental analysis results, “*In order to check the validity of the Cu(ABDC)·DMF formula, we performed elemental analysis by a combination of inductively coupled plasma atomic emission spectrometry (ICP-AES) and C, N and H combustion methods, the weight percent is presented in the table. Taking into account of the formula CuC₁₁N₂O₅H₁₂, we can obtain the following elemental ratios according to the experiment results: C/H≈11/12.4, C/N≈11/2, C/Cu≈11/1, C/O≈11/5, which is basically consistent with the theoretical value. Therefore, the element analysis finally validates a chemical formula as Cu(ABDC)·DMF for the layered semiconducting MOF that we have synthesized. We also performed elemental analysis for LS-Cu-ABDC MOFs. Taking into account of the corresponding formula for 1.4%- and 2.7%-LS-Cu-ABDC MOFs according to the results of FT-IR analysis, we can obtain the following elemental ratios according to the experiment results: C/H≈10.75/12.59, C/N≈10.75/1.84, C/O≈10.75/5.1, C/Cu≈10.75/1 for 1.4%-LS-Cu-ABDC, and C/H≈10.58/12.20, C/N≈10.58/1.90, C/O≈10.58/4.9, C/Cu≈10.58/1 for 2.7%-LS-Cu-ABDC, which is basically consistent with the following chemical formula: Cu(ABDC)_{0.75}·(BA)_{0.25}·DMF·OH_{0.25} and Cu(ABDC)_{0.58}·(BA)_{0.42}·DMF·OH_{0.42}. Therefore, all above results indicate that the chemical formula for Cu-ABDC, 1.4%- and 2.7%-LS-Cu-ABDC can be estimated to be Cu(ABDC)·DMF, Cu(ABDC)_{0.75}·(BA)_{0.25}·DMF·OH_{0.25} and Cu(ABDC)_{0.58}·(BA)_{0.42}·DMF·OH_{0.42}, respectively. Besides, the atomic ratio of C/Cu is in agreement with the theoretical results, indicating no excess Cu. Therefore, there is also no CuO in our samples.*” **have been added** in Supplementary Table 1 in page 30 of the Supplementary Information, and the original description in Supplementary Figure 3 in page 6 of the Supplementary Information, “*Considering that the presence of the distinct sequences of functionalities along the MOF backbone will cause an abundant pore environment, and will not change the crystal structure of the MOF.^{1,2} Therefore, we synthesized the Cu-ABDC using 2-amino-1,4-benzenedicarboxylic acid (ABDC) and Cu(NO₃)₂·3H₂O, and the crystal structure is exactly consistent with that of well-known Cu(tpa)·DMF.³ We also have carried out the Rietveld refined XRD patterns of the Cu-ABDC MOF using the software Rietica. As seen from Figure S3, all peaks of the MOF are indexed to the standard structure Cu(tpa)·DMF MOF with space group C 2/m (CCDC-687690), which is synthesized using 1,4-benzenedicarboxylic acid (BDC) and Cu(NO₃)₂·3H₂O and has been reported by Carson et al.¹⁻⁵ suggesting the Rietveld refinement of the XRD data reveals an almost single-phase nature of the MOF. The lattice parameters of the MOF obtained from the refinement are a = 11.1345 Å, b = 14.2439 Å and c = 7.8529 Å, which is basically in agreement with the value reported by Carson et al., indicating a reliable quality for our sample*” **has been changed to** “*Considering that the presence of the distinct sequences of functionalities along the MOF backbone will*

cause an abundant pore environment, and will not change the crystal structure of the MOF. Therefore, we synthesized the Cu-ABDC using 2-amino-1,4-benzenedicarboxylic acid (ABDC) and $\text{Cu}(\text{NO}_3)_2 \cdot 3\text{H}_2\text{O}$, and the crystal structure is exactly consistent with that of well-known monoclinic $\text{Cu}(\text{tpa})\text{-DMF}$. We also have carried out the Rietveld refined XRD patterns of the Cu-ABDC MOF using the software Rietica. As seen from Figure S3, all peaks of the MOF are indexed to the standard structure $\text{Cu}(\text{tpa})\text{-DMF}$ MOF with space group $C 2/m$ (CCDC-687690), which is synthesized using 1,4-benzenedicarboxylic acid (BDC) and $\text{Cu}(\text{NO}_3)_2 \cdot 3\text{H}_2\text{O}$ and has been reported by Carson et al., suggesting that the Rietveld refinement of the XRD data reveals an almost single-phase nature of the MOF. The lattice parameters of the MOF obtained from the refinement are $a = 11.1345 \text{ \AA}$, $b = 14.2439 \text{ \AA}$ and $c = 7.8529 \text{ \AA}$, which is basically in agreement with the value reported by Carson et al., indicating a reliable quality for our sample. Additionally, in order to exclude the Cu-related phase in our samples, synchrotron radiation powder X-ray diffraction patterns of 1.4%-LS-Cu-ABDC and the reference sample CuO were also performed. Compared to the characteristic peak at about 35.5° and 38.7° for CuO, there are no obvious peaks at the same position for 1.4%-Cu-ABDC sample (Figure S3b), indicating the absence of CuO, which can further confirm that ferromagnetic is intrinsic in our samples.” The results and description of solid-state cyclic voltammetry measurements, “The CV curves were measured at 0.1 V/s scan rate in 0.1 M TBAPF₆/DMF (TBAPF₆, Tetrabutylammonium Hexafluorophosphate) and KCl solutions. As shown in Figure S4a, the peaks at about -0.37 and -0.47 V for Cu-MOFs, and peaks at about -0.21 and -0.47 V for CuO were observed in 0.1 M TBAPF₆/DMF solution, indicating the successfully synthesized of Cu-MOFs and the absence of CuO. In 0.1 M KCl solution, two pairs of redox peaks were observed for Cu-ABDC MOFs, and only one pair of redox peaks at about 0.12 and -0.13 V for CuO. These different peak positions between CuO and Cu-MOFs indicate the absence of CuO in our samples.” **has been added** in Supplementary Figure 4 in page 7 of the Supplementary Information, respectively.

4. Question: The authors attribute the lattice expansion to the missing linker based on XRD, TEM, etc. While I agree with the interpretation, the authors need to clearly explain how the missing linker causes lattice expansion rather than contraction.

Reply: This is a good question. The Cu-ABDC MOF sample is based on a paddle-wheel Cu^{2+} dimer unit connected by four dicarboxylate moieties, which are held together by strong carboxylate-metal bonds [Kitagawa et al., *Angew. Chem. Int. Ed.* **2004**, *43*, 2334-2375; Shekhah et al., *Angew. Chem. Int. Ed.* **2009**, *48*, 5038]. After the substitution of ligands to introduce lattice strain, the bond of the dicarboxylic acid group to the dimer is broken, and the repulsive force becomes stronger between Cu ions within

a dimer. Thus, the distance increases between Cu ions within a Cu dimer, which leads to local lattice expansion, corresponding to the structure optimization results in the theoretical calculations (Supplementary Table 4) and the previous reports [*Huang et al., Nat. Commun.* **2019**, *10*, 2779; *Zhou et al., Adv. Mater.* **2021**, *33*, 2104341].

Figure R2. Schematic illustration of the changes of Cu ions distance within a dimer after lattice stress is introduced. The axial organic solve molecules are omitted for clarity in all structure models.

5. Question: Fit parameters obtained from the Rietveld refinement need to be presented in the SI.

Reply: Thank you for your suggestion. Based on the Rietveld refined XRD pattern results, we have presented the fitted parameters in **Supplementary Table 2** in page 31 of the revised Supplementary Information text.

Supplementary Table 2. The Rietveld refined XRD pattern results. The lattice parameters of the MOF obtained from the refinement are basically in agreement with the reported values, indicating a reliable quality for our samples.

Parameters	Standard	Refined
a (Å)	11.41(4)	11.13(4)
b (Å)	14.27(8)	14.24(3)
c (Å)	7.78(0)	7.85(2)
α (°)	90	90
β (°)	108.12 (9)	110.99(8)
γ (°)	90	90
R_p (%)	-	1.38
R_{wp} (%)	-	1.97

Accordingly, the results and description “**Supplementary Table 2. The Rietveld refined XRD pattern results. The lattice parameters of the MOF obtained from the refinement are basically in agreement with the reported values, indicating a reliable quality for our samples.**” has been added in Supplementary Table 2 in page 31 of the Supplementary Information,

6. Question: The authors need to provide a better explanation for the reduced saturation magnetization. Authors suggest this is due to the increased occupation of Cu 3d states. However, they disregard the possibility of copper(I) ions and suggest that all copper ions exist in the 2+ oxidation state. I find these claims contradictory. Ferromagnetically coupled copper(II) ions ($S = 1/2$) should have a saturation magnetization value of $\sim 1\mu_B$. Observed moments for all samples are significantly lower than expected. Furthermore, magnetization in 2.7%-LS-Cu-ABDC sample doesn't seem to have saturated at 10000 Oe and decreases with the increasing field above ~ 5000 Oe, which needs a clear explanation.

Reply: (1) After the substitution of ligands to introduce lattice strain, the coordination number of Cu-O remains unchanged because the defective sites of Cu ions are directly connected to hydroxide ions to balance charge [Chen *et al.*, *J. Am. Chem. Soc.* **2000**, *122*, 11559-11560; Marx *et al.*, *J. Catal.* **2011**, *281*, 76-87; Fang *et al.*, *J. Am. Chem. Soc.* **2014**, *136*, 9627-9636; Zhan *et al.*, *Adv. Funct. Mater.* **2019**, *29*, 1806720; Huang *et al.*, *Nat. Commun.* **2019**, *10*, 2779; Zhou *et al.*, *Adv. Mater.* **2021**, *33*, 2104341]. Therefore, the Cu ions still remain +2 valence, and only electron transfer (because the electron-withdrawing ability of the -OH group is lower than -(OCO)- group) and lattice swelling occur, which can be confirmed by our XPS, Cu-L edge XAS and Cu-K edge XAFS results.

On one hand, the exchange interactions after the ligand replacement are usually weaker than that for four -(OCO)- bridges, and the strength of the interaction decreases with the number of intervening bonded atoms between the moment carriers [Kurmoov *et al.*, *Chem. Soc. Rev.* **2009**, *38*, 1353-1379; Kurmoov *et al.*, *New J. Chem.*, **1998**, 1515-1524]. On the other hand, the substitution of ligands is hard to precisely control and hence not uniform during sample preparation. It is possible that antiferromagnetic coupling remains in some Cu dimers whose ligands are not replaced, and their interactions result in the "spin canting effect" and a decrease in the net magnetic moment, so the overall magnetic moment is smaller than $1\mu_B/\text{Cu atom}$ [Jin *et al.*, *Adv. Funct. Mater.* **2023**, 2214273], in agreement with our theoretical calculations results.

(2) The magnetization in 2.7%-LS-Cu-ABDC sample doesn't seem to have saturated at 10000 Oe and decreases with the increasing field above ~ 5000 Oe. These may be caused by incomplete subtraction of the background signal. We have retested and revised the *M-T* and *M-H* curves of 2.7%-LS-Cu-ABDC. To show *M-H* curves more clearly, we subtracted the paramagnetic (PM) and diamagnetic (DM) background signals by the following processes: we first acquired the PM and DM magnetization by

fitting the M - H curves measured at different temperatures in the positive and negative field regions. Then we obtained the field-dependent PM and DM magnetization ($M_{P+D}(H)$) relationship with the applied field by the least square method: $M_{P+D}(H) = a \cdot H \pm b$ where a and b are the fitting parameters. Taking the M - H curves at 300 K of 2.7%-LS-Cu-ABDC sample as an example, the obtained $M_{P+D}(H)$ curve is shown in **Figure R3**, where $a = 9.02 \times 10^{-5}$, $b = 6.86 \times 10^{-2}$. After that, the M - H curves of our samples can be extracted by subtracting the PM+DM signals from the raw data, as shown in **Supplementary Figure 20b**.

Figure 4c. M - T curves of the 2.7%-LS-Cu-MOFs.

Figure R3. The M - H curves before and after subtracting the PM+DM signals.

Supplementary Figure 20b. Magnetic hysteresis loops of the 2.7%-LS-Cu-MOFs.

Accordingly, the original description in page 9 of the main text, “*It is noting that the magnetic moment is smaller than $1\mu_B/\text{Cu}$ atom. This is due to that the substitution ligands are not perfectly uniform within the sample, which will cause antiferromagnetic coupling remains in some Cu dimers whose ligands are not replaced.*” **has been changed to** “*It is noting that the magnetic moment is smaller than $1\mu_B/\text{Cu}$ atom and the saturation magnetization and coercivity are both decreased when the lattice strain increases from 1.4% to 2.7%. This is due to the increased occupation of Cu 3d states. On one hand, the exchange interactions after the ligand replacement are usually weaker than that for four -(OCO)- bridges, and the strength of the interaction decreases with the number of intervening bonded atoms between the moment carriers. On the other hand, the substitution of ligands is hard to precisely control and hence not uniform during sample preparation. It is possible that antiferromagnetic coupling remains in some Cu dimers whose ligands are not replaced, and their interactions result in the “spin canting effect” and a decrease in the net magnetic moment, so the overall magnetic moment is smaller than $1\mu_B/\text{Cu}$ atom, in agreement with our theoretical calculations results.*”

7. Question: The authors need to clearly state whether their samples exhibit long-ranged or short-ranged magnetic order. The missing linkers should reduce the magnetic correlation within the system, and benzene carboxylates do not provide strong super-exchange interaction between the magnetic centers. Any magnetic order, if present as the authors claim, should be short-ranged. Under these assumptions, I’m skeptical about the use of RSG phenomenon to describe the magnetic properties of their samples as the term is applicable for systems that exhibit long-range magnetic order. As such, I strongly suggest finding an alternative interpretation of the magnetic properties. If the authors believe their samples exhibit long-ranged ferromagnetic order, proper references and additional experimental data must be provided. Some suggestions include out-of-phase AC magnetic susceptibility data, neutron diffraction, computational results on a larger unit-cell, etc.

Reply: Thank you for your comment. The Cu-ABDC MOF sample is based on a paddle-wheel Cu^{2+} dimer unit connected by four dicarboxylate moieties, which are held together by strong carboxylate-metal bonds [*Shekhah et al., Angew. Chem. Int. Ed.* **2009**, *48*, 5038]. According to the previous reports, the strong super-exchange coupling between two Cu ions within a dimer has prominent antiferromagnetic (AFM) spin

alignment without any macroscopic magnetic moment, and weaker ferromagnetic (FM) coupling between dimers via itinerant π electrons in the organic linkers, indicating Cu dimers are coupled to each other via itinerant π electrons in the organic linkers to provide long-range magnetic order [Hiberty *et al.*, *J. Am. Chem. Soc.* **1995**, *117*, 9875-9880; Zhang *et al.*, *J. Appl. Phys.* **2000**, *87*, 6007-6009; Moulton *et al.*, *Angew. Chem. Int. Ed.* **2002**, *41*, 2821-2824; Kurmoo *et al.*, *Chem. Soc. Rev.* **2009**, *38*, 1353-1379; Shen *et al.*, *J. Am. Chem. Soc.* **2012**, *134*, 17286-17290; Tiana *et al.*, *Chem. Commun.* **2014**, *50*, 13990-13993; Gu *et al.*, *Appl. Phys. Lett.* **2015**, *107*, 183301]. After the cleavage of the ligand, only part of the ABDC linkers are replaced. Therefore, these Cu dimers can always be coupled via itinerant π electrons in the remaining organic linkers to provide long-range magnetic order (**Figure R2 and Supplementary Figure 26**), though the exchange interactions are weaker than that for four -(OCO)- bridges [Kurmoo *et al.*, *Chem. Soc. Rev.* **2009**, *38*, 1353-1379]. Besides, we also performed AC magnetic susceptibility, neutron diffraction measurements and theoretical calculations on a larger unit-cell, and all the results are summarized below:

(1) AC magnetic susceptibility measurements. In order to further investigate the long-range ferromagnetic order, we performed the in-phase and out-of-phase AC magnetic susceptibility $\chi'(T)$ and $\chi''(T)$ for 1.4%- and 2.7%-LS-Cu-ABDC MOF as a function of temperature at an AC field of $H_{ac} = 2$ Oe and $H_{dc} = 0$ Oe with several frequencies ($f = 1, 10, 100, 500, \text{ and } 1000$ Hz) at the temperature range of 50-360 K. However, the $\chi'(T)$ curves have no obvious changes with the frequency, and the out of phase χ'' (absorption) remains practically zero. These results indicate no obvious short-range order signal in our samples, such as spin glass behavior [Park *et al.*, *Nat. Chem.* **2021**, *13*, 594-598; Mustonen *et al.*, *Nat. Commun.* **2018**, *9*, 1085; Singh *et al.*, *Phys. Rev. B* **2008**, *77*, 144403]. Additionally, it is possible that the transition temperature has not been reached or the response of our samples to AC magnetic susceptibility is relatively weak, causing no obvious signal in out-of-phase AC magnetic susceptibility.

Supplementary Figure 21. Temperature variation of the AC susceptibility measurement for 1.4% and 2.7%-LS-Cu-ABDC MOF. a-b, in-phase and out-of-phase AC magnetic susceptibility measurement for 1.4%-LS-Cu-ABDC at various frequencies from 1 Hz to 1000 Hz with $H_{ac} = 2$ Oe and $H_{dc} = 0$ Oe from 50 to 360 K. **c-d**, in-phase and out-of-phase AC magnetic susceptibility measurement for 2.7%-LS-Cu-ABDC at various frequencies from 1 Hz to 1000 Hz with $H_{ac} = 2$ Oe and $H_{dc} = 0$ Oe from 50 to 360 K.

(2) Theoretical calculation results on a larger unit-cell. To confirm the long-range ferromagnetic order in our samples, theoretical calculation results on a larger unit-cell were performed. The spatial distribution of spin densities on a large unit-cell was shown in **Supplementary Figure 26**. We can clearly see that the spin states are present in the benzene ring of the organic linkers, indicating that the delocalized π electrons in the organic linkers provide the bridge for the exchange interaction between Cu dimers. After the substitution of ligands, the spatial distribution of spin densities on the organic linkers is always present, and just gets weaker with the cleavage of the linkers. We also use 0.0002 spins per bohr³ to show the spatial distribution of spin densities in **Figure R4** (in order to exhibit the spin densities of Cu dimers more clearly, we use 0.002 spins per bohr³ in Fig. 4e-h in the main text). It also exhibits the spin states in the benzene ring of the organic linkers, indicating the long-range ferromagnetic order. And the spin densities of the axial solvent molecules and partial atoms are omitted for clarity in all structure models.

Supplementary Figure 26. Theoretical calculation results on a larger unit-cell for 1.4%-LS-Cu-ABDC (a) and 2.7%-LS-Cu-ABDC with the adjacent ligands (b), respectively. Red and blue iso-surfaces represent positive and negative spin densities, and the value is 0.0002 spins per bohr³, Copper, carbon, nitrogen, oxygen and hydrogen atoms are shown in blue, gray, yellow, red and white, respectively, and the axial solvent molecular and partial atoms are omitted for clarity in all structure models.

Figure R4. Electronic structure calculations of Cu-MOFs. e-h Corresponding spin density iso-surface distribution of a Cu dimer supercell for Cu-ABDC at AFM state (e), 1.4%-LS-Cu-ABDC at FM state (f), 2.7%-LS-Cu-ABDC with the adjacent (g) and diagonal (h) ligands at FM state, respectively. Red and blue iso-surfaces represent positive and negative spin densities, and the value is 0.0002 spins per bohr³, and the axial solvent molecular are omitted for clarity in all structure models.

(3) Neutron diffraction analysis. Neutrons have magnetic moments, which can interact with atomic magnetic moments to produce neutron-specific magnetic diffraction, so neutron diffraction is an extremely important means to study magnetic structures [Zhou *et al.*, *Phys. Rev. Lett.* **2011**, 106, 147204; Hagihala *et al.*, *npj Quantum Mater.* **2019**, 4, 14; Kozlenko *et al.*, *npj Quantum Mater.* **2021**, 6, 19]. We have tried neutron diffraction measurements at the beamline of the general-purpose powder diffractometer (GPPD) of the China Spallation Neutron Source (CSNS). However, the H atoms on the benzene ring of Cu-MOF cannot be fully deuterated, resulting in a large background and no obvious signal peak. Therefore, it needs further

research to obtain the information of crystal and magnetic structure by fully deuterating or performing low-temperature experiments. Investigation of the magnetic structure and domain size in MOFs based on paddle-wheel units presents an intriguing area for further study.

Accordingly, the original data and description in Supplementary Figure 17 in page 20 of the Supplementary Information, “*In order to further investigate the possibility of glassy state, we performed the real part of AC magnetic susceptibility $\chi'(T)$ (dispersion) for 2.7%-LS-Cu-ABDC MOF as a function of temperature at an AC field of $H_{ac} = 2$ Oe and $H_{dc} = 0$ Oe with several frequencies ($f = 1, 10, 100, 500,$ and 1000 Hz) at the temperature range of 100-360 K. However, the peak positions at about 275 K have no obvious changes with the frequency. Moreover, the imaginary part χ'' (absorption) remains practically zero. On the other hand, we also measured the real part of AC magnetic susceptibility $\chi'(T)$ for 2.7%-LS-Cu-ABDC MOF as a function of temperature at an AC field of $H_{ac} = 2$ Oe with several DC field ($H_{dc} = 0, 10, 20, 50, 100$ and 200 Oe) at temperature range 100-360 K under $f = 10$ and 500 Hz, respectively. The peak positions also remain unchanged at different DC fields. These results indicate no obvious spin glass signal in our samples. The above results lead to the conclusion of complete absence of spin glass state.” **has been changed to** Supplementary Figure 21 in page 24 “*In order to further investigate the long-range ferromagnetic order, we performed the in-phase and out-of-phase AC magnetic susceptibility $\chi'(T)$ and $\chi''(T)$ for 1.4%- and 2.7%-LS-Cu-ABDC MOF as a function of temperature at an AC field of $H_{ac} = 2$ Oe and $H_{dc} = 0$ Oe with several frequencies ($f = 1, 10, 100, 500,$ and 1000 Hz) at the temperature range of 50-360 K. However, the $\chi'(T)$ curves have no obvious changes with the frequency, and the out-of-phase χ'' (absorption) remains practically zero. These results indicate no obvious short-range order signal in our samples, such as spin glass behavior. Additionally, it may be that the transition temperature has not been reached or the response of our samples to AC magnetic susceptibility is relatively weak, causing no obviously signal in out-of-phase AC magnetic susceptibility.” and the spatial distribution of spin densities results on a large computational unit-cell and description “*To confirm long-range ferromagnetic order in our samples, theoretical calculation results on a large unit-cell were performed. The spatial distribution of spin densities on a larger unit-cell was shown in Figure S26. We can clearly see that the spin states are present in the benzene ring of the organic linkers, indicating the delocalized π electrons in the organic linkers provide the bridge for the exchange***

interaction between Cu dimers. After the substitution of ligands, the spatial distribution of spin densities on the organic linkers is always present, and just gets weaker with the cleavage of the linkers. And the spin densities of the axial solvent molecules and partial atoms are omitted for clarity in all structure models.” has been added in Supplementary Figure 26 in page 29 of the Supplementary Information text.

8. Question: It is unclear which features in the TDOS plots at the Fermi level suggest strong hybridization of the Cu and ligand orbitals. Authors need to clearly state contributions from individual atoms and whether these contributing orbitals have π symmetry. Otherwise, I suggest deleting the sentence: “The TDOSs near the Fermi level...”

Reply: Thank you for your professional suggestions. In order clearly exhibit contributions from individual atoms, we have replotted **Supplementary Figure 22** in page 25 in the revised Supplementary Information text. Our samples are based on paddle-wheel units formed by attaching 4 dicarboxylate moieties to Cu^{2+} dimers, yielding planar sheets with 4-fold symmetry [Shekhah *et al.*, *Angew. Chem. Int. Ed.* **2009**, *48*, 5038]. After introducing ligand cleavage in Cu-ABDC, the TDOSs (see details in **Supplementary Figure 22**) near the Fermi level of LS-Cu-ABDC display increased hybridization of the orbitals from Cu($3d$), C($2p$) and O($2p$), indicating the ferromagnetic order.

Supplementary Figure 22. Theoretical calculation. The calculated densities of states (DOS) for Cu-ABDC (**a**), 1.4%-LS-Cu-ABDC (**b**) and 2.7%-LS-Cu-ABDC (**c-d**), respectively. DFT calculations reveal that the long-range ferromagnetic interaction between adjacent dimers mainly arises from the hybridization between Cu $3d$, O $2p$ and C $2p$ orbitals.

(2) In order to further confirm the π symmetry, we also performed the calculation of molecular orbital. The calculated partial densities of states (PDOS) and corresponding spatial distribution of one of the molecular orbitals with π symmetry for 1.4%-LS-Cu-ABDC were shown in **Supplementary Figures 23 and 24**. The PDOS (**Supplementary Figure 23**) near the Fermi level of 1.4%-LS-Cu-ABDC display clear hybridization of the orbitals from Cu($3d$) and C($2p_z$), and spatial distribution of one of the molecular orbitals with π symmetry (**Supplementary Figure 24**) clearly shows the hybridization between the orbitals from C($2p_z$) and Cu($3d$), indicating the presence of delocalized π electrons and π symmetry.

Supplementary Figure 23. The calculated partial densities of states (PDOS) for 1.4%-LS-Cu-ABDC. The PDOS near the Fermi level of 1.4%-LS-Cu-ABDC display increased hybridization of the orbitals from Cu($3d$) and C($2p_z$), indicating the presence of delocalized π electrons and π symmetry.

Supplementary Figure 24. Spatial distribution of one of the molecular orbitals with π symmetry for 1.4%-LS-Cu-ABDC. It displays clear hybridization between the orbitals from C($2p_z$) and Cu($3d$), indicating the presence of delocalized π electrons and π symmetry. Yellow iso-surfaces represent the spatial distribution of the C($2p_z$) orbital. Copper, carbon, nitrogen, oxygen and hydrogen atoms are shown in blue, gray, yellow, red and white, respectively, and the axial solvent molecules are omitted for clarity.

Accordingly, the calculated partial densities of states (PDOS) “*The calculated partial densities of states (PDOS) for 1.4%-LS-Cu-ABDC. The PDOS near the Fermi level of 1.4%-LS-Cu-ABDC display increased hybridization of the orbitals from Cu($3d$) and C($2p_z$), indicating the presence of delocalized π electrons and π symmetry.*” and “*Spatial distribution of one of the molecular orbitals with π symmetry for 1.4%-LS-Cu-ABDC. It displays clear hybridization between the orbitals from C($2p_z$) and Cu($3d$), indicating the presence of delocalized π electrons and π symmetry. Yellow iso-surfaces represent the spatial distribution of the C($2p_z$) orbital. Copper, carbon, nitrogen, oxygen and hydrogen atoms are shown in blue, gray, yellow, red and white, respectively, and the axial solvent molecules are omitted for clarity.*” has been added in Supplementary Figures 23 and 24 in pages 26-27 of the Supplementary Information text.

9. Question: I’m skeptical about the statement: “Therefore, the indirect exchange interaction between the localized Cu spins...”. The manuscript lacks the experimental evidence to support the delocalized π electrons. As such, I suggest removing the sentence.

Reply: Thank you for your nice question. To auxiliary confirm the delocalized π electrons, we carried out the ultraviolet-visible-near infrared (UV-Vis-

NIR) spectroscopy and C *K*-edge X-ray absorption spectroscopy (XAS). The results are summarized below:

(1) UV-Vis-NIR spectroscopy measurement. The UV-Vis-NIR spectra of our samples and the ligand 2-amino-1, 4-benzenedicarboxylic acid (ABDC) are shown in **Supplementary Figure 17**. Remarkably, the UV-Vis-NIR spectra for all the Cu-ABDC samples exhibit absorption at about 13609 cm^{-1} extending to the near-infrared (NIR) region. However, this absorption band was not found in the ABDC linker, indicating a strong *d*- π conjugation between the metal node and the organic linkers, which means the existence of exchange interaction between the metal nodes and the organic linkers [Dong *et al.*, *Nat. Commun.* **2018**, *9*, 2637; Pham *et al.*, *J. Am. Chem. Soc.* **2022**, *144*, 23, 10615-10621].

Supplementary Figure 17. UV-Vis-NIR spectra of Cu-MOFs. UV-Vis-NIR spectra of Cu-MOFs and ABDC linker (a). Tauc plot derived from absorption of Cu-MOFs (b).

(2) C *K*-edge XAS analysis. The C *K*-edge XAS spectra (**Supplementary Figure 25**) of Cu-ABDC MOFs exhibit four distinct absorption peaks. A domain peak A at about 285.3 eV is attributed to C-C π^* (ring) excitations, and peak B (286.4 eV), peak C (287.7 eV) and peak D (288.9 eV) can be ascribed to the functional groups containing O and N [Liang *et al.*, *Nat. Mater.* **2011**, *10*, 780-786; Liang *et al.*, *J. Am. Chem. Soc.* **2012**, *134*, 3517-3523; Wang *et al.*, *Energy Environ. Sci.* **2013**, *6*, 2900-2906; Wang *et al.*, *J. Am. Chem. Soc.* **2017**, *139*, 9419-9422; Han *et al.*, *Nat. Commun.* **2020**, *11*, 2209; Hu *et al.*, *Nat. Commun.* **2021**, *12*, 1854]. Specifically, the photon energy position of π^* in our samples is in accordance with that of HOPG, indicating electronic transitions from C 1s core level to delocalized π^* energy level in Cu-ABDC MOFs [Tang *et al.*, *Appl. Phys. Lett.* **2001**, *79*, 3773-3775; Yazyev, *et al.*, *Phys. Rev. B* **2007**, *75*, 125408-125411; Guillen *et al.*, *Phys. Rev. Lett.* **2006**, *96*, 107203-107206; Palacios *et al.*, *Phys. Rev. B* **2008**, *77*, 195428-195441; Pan *et al.*, *Nature* **2023**, *614*, 95-101]. Therefore, we

can conclude that delocalized π electrons are present in our MOFs, and the long-range ferromagnetic order is preferred due to the coupling between localized spin states via delocalized π electrons in the organic linkers.

Supplementary Figure 25. C K-edge XAS spectra of Cu-ABDC MOFs and the reference sample HOPG.

Accordingly, the description in Supplementary Figure 6 in page 7 of the Supplementary Information text “Two absorption peaks D (21453 cm^{-1}) and E (13609 cm^{-1}) can be detected, which are attributed to charge transfer between ligand and metal and metal-radical spin-exchange originated from d-d transition.¹³ There also present three peaks A (45610 cm^{-1}), B (38021 cm^{-1}) and C (27934 cm^{-1}), which may be ascribed to the transition of the $\pi\rightarrow\pi^*$ of the organic functional groups, the transition of the $\pi\rightarrow\pi^*$ of the aromatic rings and the electron transfer transition from ligand to metal.^{12, 14, 15} Moreover, bandgaps are fitted as 1.20, 1.22 and 1.27 eV for Cu-ABDC, 1.4%-LS-Cu-ABDC and 2.7%-LS-Cu-ABDC, respectively, suggesting its semiconductor behaviors and the bandgaps gradually increase with the ligand cleavages, in agreement with theoretical calculation results below.” **has been changed to** Supplementary Figure 17 in page 20 “Two absorption peaks D (21453 cm^{-1}) and E (13609 cm^{-1}) can be detected, which are attributed to charge transfer between ligand and metal and metal-radical spin-exchange originated from d-d transition. There also present three peaks A (45610 cm^{-1}), B (38021 cm^{-1}) and C (27934 cm^{-1}), which may be ascribed to the transition of the $\pi\rightarrow\pi^*$ of the organic functional groups, the transition of the $\pi\rightarrow\pi^*$ of the aromatic rings and the electron transfer transition from ligand to metal. Moreover, bandgaps are fitted as 1.20, 1.22 and 1.27 eV for Cu-ABDC, 1.4%-LS-Cu-ABDC and 2.7%-LS-Cu-ABDC, respectively, suggesting its semiconductor behaviors and the bandgaps gradually increase with the ligand cleavages, in agreement with theoretical calculation results below. Remarkably, the UV-Vis-NIR spectra for all the Cu-ABDC samples

exhibit absorption at about 13609 cm^{-1} extending to the near-infrared (NIR) region. However, this absorption band was not found in the ABDC linker, indicating a strong $d-\pi$ conjugation between the metal node and the organic linkers, which means the existence of exchange interaction between the metal nodes and the organic linkers.” and the C K-edge XAS spectra and description “The C K-edge XAS spectra of Cu-ABDC MOFs exhibit four distinct absorption peaks. A domain peak A at about 285.3 eV is attributed to C-C π^* (ring) excitations, and peak B (286.4 eV), peak C (287.7 eV) and peak D (288.9 eV) can be ascribed to the functional groups containing O and N. Specifically, the photon energy position of π^* in our samples is in accordance with that of HOPG, indicating electronic transitions from C 1s core level to delocalized π^* energy level in Cu-ABDC MOFs. Therefore, we can conclude that delocalized π electrons are present in our MOFs, and the long-range ferromagnetic order is preferred due to the coupling between localized spin states via delocalized π electrons in the organic linkers.” **has been added** in Supplementary Figure 25 in page 28 of the Supplementary Information text.

10. Question: The authors need to clearly describe experimental procedures for all of their material characterization, including Mott-Schottky analysis and structure determination in Fig S1.

Reply: Thank you for your suggestion. The structure determination and experimental procedures of Mott-Schottky, magnetic measurements and X-ray Absorption Structure (XAS), and conductivity measurements have been summarized below:

(1) Structure determination. The presence of the distinct sequences of functionalities along the MOF backbone will cause an abundant pore environment, and not change the crystal structure of the MOFs [Deng *et al.*, *Science* **2010**, 327,846-850; Matthew *et al.*, *Chem. Commun.* **2001**, 2532-2533]. Therefore, we synthesized the Cu-ABDC using 2-amino-1,4-benzenedicarboxylic acid (ABDC) and $\text{Cu}(\text{NO}_3)_2 \cdot 3\text{H}_2\text{O}$, and the crystal structure is exactly consistent with that of well-known $\text{Cu}(\text{tpa}) \cdot \text{DMF}$, which is synthesized using 1,4-benzenedicarboxylic acid (BDC) and $\text{Cu}(\text{NO}_3)_2 \cdot 3\text{H}_2\text{O}$ [Carson *et al.*, *Eur. J. Inorg. Chem.* **2009**, 16, 2338-2343; Tania *et al.*, *Nat. Mater.* **2015**, 14, 48-55; Zhan *et al.*, *Adv. Funct. Mater.* **2019**, 29, 1806720]. Then, we obtained the crystal structure data from the Cambridge Crystallographic Data Centre (CCDC-687690), which has been reported by Carson *et al.* [Carson *et al.*, *Eur. J. Inorg. Chem.* **2009**, 16, 2338-2343]. The crystal structure of the $\text{Cu}(\text{tpa}) \cdot \text{DMF}$ is monoclinic with space group $C2/m$ (NO. 12). The lattice parameters are $a = 11.4143\text{ \AA}$, $b = 14.2687\text{ \AA}$, $c = 7.7800$

\AA , $\beta=108.119^\circ$, $V=1204.27 \text{ \AA}^3$, respectively. The diffraction patterns of our Cu-ABDC MOF match well with that of Cu(tpa)·DMF MOF, indicating the same crystal structure. Besides, for structure characterizations of the Cu-MOFs, the spin-polarized DFT calculations implemented in the Quantum Espresso software package were performed [Giannozzi et al., *J. Phys. Condens. Matter* **2009**, *21*, 395502]. The electron-ion interaction and electron exchange-correlation was described with the projected augmented wave (PAW) method with a kinetic energy cutoff of 1020 eV and generalized gradient approximation (GGA) in the Perdew-Burke-Ernzerhof (PBE) parametrization. Then the DFT-D3 scheme was employed to process the long-range van der Waals interaction. The Brillouin zone was integrated with a $3 \times 2 \times 4$ k-grid. The convergences for self-consistent field calculations and atomic structure optimizations were 2×10^{-7} eV/atom and 0.05 eV/ \AA . Some calculation results were visualized with VMD software [Humphrey et al., *J. Mol. Graphics*. 1996, *14*, 33-38].

(2) Electrochemical measurements. Mott-Schottky and solid-state cyclic voltammetry measurements were performed on a CHI760D electrochemical workstation using a typical three-electrode system. The Cu-ABDC nanosheets on glassy carbon electrodes acted as the working electrode with a platinum mesh as the counter electrode and an Ag/AgCl reference electrode. Typically, 4 mg of samples and 30 μL Nafion solution (5 wt%, Sigma Aldrich) were dispersed in 1 mL ethanol solution to form a homogeneous ink assisted by ultrasonic method. Mott-Schottky measurements were conducted in 1 M KOH (aq) electrolytes continuously purged with 99.999% N_2 (Praxair) and at a sweep rate of 5 mV/s. Then, Mott-Schottky analysis was carried out in the linear region of the C^{-2} curve from 0.9 to 1.3 V vs Ag/AgCl with a frequency of 2 kHz. The solid-state cyclic voltammetry was measured at 0.1 V/s scan rates in 0.1 M TBAPF₆/DMF and KCl solutions.

(3) Magnetic measurements. Magnetic measurements were performed using a SQUID (Superconducting Quantum Interference Design). Variable-temperature direct current (d.c.) magnetic susceptibility of the samples was measured using a quartz sample holder in zero-field cooling and field cooling sequence with the applied magnetic field of 500 Oe with a temperature range of 5-300 K. In our magnetic characterizations, M - H curves were measured at the scan rate of the magnetic field 50 Oe/s in the range of 0-300 Oe, 200 Oe/s in the range of 400-3000 Oe and 2000 Oe/s in the range of 4000-10000 Oe with the field up to 2 T, M - T curves were measured over a temperature range of 5-300 K at 500 Oe with the heating rate of 10 ~ 15 K/min. Variable-temperature alternating current (a.c.) magnetic susceptibility of the samples was measured at the temperature

range of 50-360 K, and at an AC field of $H_{ac} = 2$ Oe with DC field $H_{dc} = 0$ under different frequencies ($f = 1, 10, 100, 500, \text{ and } 1000$ Hz).

(4) XAS measurements. The Cu *K*-edge X-ray absorption fine structure (XAFS) data were collected in the transmission mode and calibrated using Cu foil. During the measurement, samples were placed at room temperature. Both the Cu *L*-edge and O *K*-edge were measured by means of total electron yield (TEY), measuring the drain current as a function of the photon energy. Multiple scans were measured and averaged. To get a great signal-to-noise ratio for the absorption curves for the Cu *L*-edge X-ray magnetic circular dichroism (XMCD) experiments, we performed the following procedure: Firstly, the sample holder was cooled down to the lowest possible temperature and kept for four hours to get stable temperature conditions, and XMCD measurements were carried out. Next, XMCD spectra were acquired eight times and averaged at the applied field positive and negative 0.6 T, respectively.

(5) Temperature-dependent electrical conductivity measurement. The pressed pellets were prepared by adding about 20 mg samples (heated at 100°C under argon overnight) in a 6 mm inner diameter split sleeve under the applied pressure of 20 MPa. The thickness of the pressed pellet was about 0.49 mm. No binder or conducting additive was added to the samples. Then, four-probe points were placed onto the top of the pressed pellets using conductive silver adhesive. All electrical transport measurements were based on the four-probe method using a probe station with a source meter Keithley 2614B. *I-V* curves were collected by scanning the current in the voltage range from +5 V to -5 V.

(6) Photoconductivity measurements. The pressed pellets were prepared by adding about 14 mg samples (heated at 100°C under argon overnight) in a 6 mm inner diameter split sleeve under the applied pressure of 20 MPa. The thickness of the pressed pellet was about 0.37 mm. And Au particles were deposited on the surface of the pressed pellet samples as electrodes via the sputtering method. The photo-electrical properties of the as-fabricated Cu-ABDC were characterized by a probe station using a 4200A-SCS source meter (Keithley) with Clarius software using 2-probe voltage linear scan mode. The incident light source was LED with wavelength of 1000 nm, and the light intensity was calibrated using an optical power meter (Newport Model No. 2936 R). The photoconductivity characterization was carried out in ambient conditions by applying a sweeping bias of 10 V at room temperature.

Accordingly, the structure determination, photoconductivity measurements, Mott-

Schottky measurements and solid-state cyclic voltammetry **has been added in** Supplementary Note 1-3 in page 2 in Supplementary Information, and the description in Supplementary Figure 7 in page 8 of the Supplementary Information text “*Mott-Schottky analysis was carried out in the linear region of the C^{-2} curve from 0.9 to 1.3 V vs Ag/AgCl with frequency of 2 kHz in 1 M KOH.^{16, 17} All the Cu-MOFs exhibit p-type semiconductor character, in agreement with theoretical calculation results, and the hole carrier density of Cu-ABDC is about 2.5 and 3.5 time larger than that of the 1.4%-LS-Cu-ABDC and 2.7%-LS-Cu-ABDC, respectively.*” **has been changed to** Supplementary Figure 18 in page 21 “*Mott-Schottky analysis was carried out in the linear region of the C^{-2} curve from 0.9 to 1.3 V vs Ag/AgCl with a frequency of 2 kHz in 1 M KOH. Typical negative slopes of the linear region in the Mott-Schottky plots can be found, indicating that all the Cu-MOFs exhibit p-type semiconductor character, in agreement with theoretical calculation results. Additionally, the carrier concentration is inversely proportional to the slope of the plots and can be calculated from the function as stated in previous report. Therefore, the hole carrier density of Cu-ABDC is about 2.5 and 3.5 times larger than that of the 1.4%-LS-Cu-ABDC and 2.7%-LS-Cu-ABDC, respectively.*”

Minor points

1. References need to be formatted properly.
2. For all data normalization performed, the authors should clearly state and explain the procedure.

Reply: Thank you for your suggestion. We have revised the format of references and the procedure of normalization has been added to the corresponding position.

Normalized procedure. For Cu L -edge XAS, firstly the linear pre-edge background is subtracted, and the L_3 and L_2 edges were normalized on intensities of the corresponding L_2 peaks in order to reveal the variation of the d -orbital occupation state more clearly. For O K -edge XAS, the linear pre-edge background was normalized to greatly exhibit the hybridization between O $2p$ orbitals and Cu $3d$; EPR spectra were also normalized using the same method. FT-IR and Raman spectra were normalized by the vibration intensity of C=C because of the same amount of the organic ligand.

Reply to Reviewer 2

Comments: In the current manuscript, Yan and coworkers demonstrate that a 2D semiconducting antiferromagnetic Cu-MOF can be endowed with intrinsic room-

temperature ferromagnetic coupling using a ligand cleavage strategy to regulate the inner magnetic interaction within the Cu dimers. The authors provide an unambiguous evidence for intrinsic ferromagnetism using the element-selective X-ray magnetic circular dichroism (XMCD) technique. Systematic characterizations as well as theoretical calculations confirm that the change of magnetic coupling is caused by the increased distance between Cu atoms within a Cu dimer. This work is interesting and might provide an effective avenue to design and fabricate MOF-based semiconducting room-temperature ferromagnetic materials. It is suitable for publication in *Nat. Commun.* after addressing the following minor points:

Reply: Thank you very much for your positive comments. We have revised the manuscript according to your advice and hope it could be published in *Nat. Commun.*.

1. Question: In order to understand the magnetic properties clearly, more information should be given. Which sample holder was used during the measurements in the SQUID? Moreover, please show the calculated details for the measured magnetic moment from emu to μ_B/Cu for the *M-H* curves.

Reply: Thank you very much for your nice suggestion. We used the straw to hold the sample during the magnetic measurements. The straw is diamagnetism and the magnetic signal is negligible compared to our samples, and the magnetic results were obtained after subtracting the straw signal.

(1) In order to reduce the impact of background signals, we also retested the magnetism using a quartz sample holder. To show *M-H* curves more clearly, we subtracted the paramagnetic (PM) and diamagnetic (DM) background signals by the following processes: we first acquired the PM and DM magnetization by fitting the *M-H* curves measured at different temperatures in the positive and negative field regions. Then we obtained the field-dependent PM and DM magnetization ($M_{\text{P+D}}(H)$) relationship with the applied field by the least square method: $M_{\text{P+D}}(H) = a * H \pm b$ where *a* and *b* are the fitting parameters. Taking the *M-H* curves at 300 K of 2.7%-LS-Cu-ABDC sample as an example, the obtained $M_{\text{P+D}}(H)$ curve is shown in **Figure R3**, where $a = 9.02 \times 10^{-5}$, $b = 6.86 \times 10^{-2}$. After that, the *M-H* curves of our samples can be extracted by subtracting the PM+DM background signals from the raw data, as shown in **Supplementary Figure 20**.

Supplementary Figure 20. The M - H curves at different temperatures for Cu-MOFs. M - H curves of Cu-ABDC (a), 1.4%-LS-Cu-ABDC (b) and 2.7%-LS-Cu-ABDC (c), respectively.

(2) For the measured magnetic moment from emu to μ_B/Cu for the M - H curves, we performed the following procedure: firstly, $1 \text{ emu} = 1.0783 \times 10^{20} \mu_B$. According to the analysis of the chemical composition of all samples, we can denote the formula weight of Cu-MOFs as M . The number of moles of Cu atoms in these samples for magnetic measurement is denoted as n , so we can obtain the number of Cu atoms in these samples during magnetic measurement $N = N_0 \times n$, in which $N_0 = 6.02 \times 10^{23}$. Therefore, the measured magnetic moment from emu to μ_B/Cu can be obtained from the function: $\text{emu} \times 1.0783 \times 10^{20} \mu_B/N$.

2. Question: As shown in Fig. 3d, the saturated magnetization of the 1.4%-LS-Cu-ABDC is much higher than that of the 2.7%-LS-Cu-ABDC, please give more discussion.

Reply: Thank you for your professional suggestions. After the substitution of ligands to introduce lattice strain, the bond between Cu ions and carboxyl group $-(\text{OCO})-$ is broken, and the defective sites of Cu ions are directly connected by hydroxide ions to balance the charge [Chen *et al.*, *J. Am. Chem. Soc.* **2000**, 122, 11559-11560; Marx *et al.*, *J. Catal.* **2011**, 281, 76-87; Fang *et al.*, *J. Am. Chem. Soc.* **2014**, 136, 9627-9636; Zhan *et al.*, *Adv. Funct. Mater.* **2019**, 29, 1806720; Huang *et al.*, *Nat. Commun.* **2019**, 10, 2779; Zhou *et al.*, *Adv. Mater.* **2021**, 33, 2104341], which causes the increased occupation of Cu 3d states, in agreement with the XAFS and XPS results. The occupation of Cu 3d states increases with the cleavage of the organic linkers. Additionally, the exchange interactions after the ligand replacement are usually weaker than that for four $-(\text{OCO})-$ bridges, the general view of the strength of the interaction, based on experimental observations and in some cases on theory, is that it decreases with the number of intervening bonded atoms between the moment carriers [Kurmoo *et*

al., Chem. Soc. Rev. **2009**, *38*, 1353-1379; Kurmoo *et al., New J. Chem.,* **1998**, 1515-1524], resulting in the decreased delocalization of the organic linkers and the increased localization of the dimers, in agreement with the theoretical calculations results (Supplementary Figure 26). Therefore, the magnetization of the 1.4%-LS-Cu-ABDC is much higher than that of the 2.7%-LS-Cu-ABDC.

Accordingly, the original description in page 9 of the main text, “*The saturation magnetization and coercivity are both decreased when the substitution of ligands increases from 1.4% to 2.7%, which may be due to the increased occupation of Cu 3d states.*” **has been changed to** “*It is noting that the magnetic moment is smaller than $1\mu_B/\text{Cu}$ atom and the saturation magnetization and coercivity are both decreased when the lattice strain increases from 1.4% to 2.7%. This is due to the increased occupation of Cu 3d states. On one hand, the exchange interactions after the ligand replacement are usually weaker than that for four -(OCO)- bridges, and the strength of the interaction decreases with the number of intervening bonded atoms between the moment carriers. On the other hand, the substitution of ligands is hard to precisely control and hence not uniform during sample preparation. It is possible that antiferromagnetic coupling remains in some Cu dimers whose ligands are not replaced, and their interactions result in the “spin canting effect” and a decrease in the net magnetic moment, so the overall magnetic moment is smaller than $1\mu_B/\text{Cu}$ atom, in agreement with our theoretical calculations results.*”

Reviewers' Comments:

Reviewer #1:

Remarks to the Author:

The authors have mostly addressed the issues with additional experimental investigation and discussions. The manuscript has been much improved. However, a few important points remain.

1. The updated manuscript still lacks experimental evidences to support long-ranged ferromagnetic order in Cu-ABDC MOFs. The AC magnetic data exhibit broad in-phase susceptibility peak and the absence of out-of-phase susceptibility peak. Further, magnetic scattering from neutron diffraction data could not be observed. In fact, computational results stand as the single support for the long-ranged ferromagnetic order. The authors should clearly indicate this limitation in their manuscript by including any statements relevant to the following: "Computational results suggests long-ranged ferromagnetic order, but further experimental investigation are needed". Specifically, while terms like "ferromagnetic coupling" may be fine, refrain from using the terms related to "long-ranged" and "magnetic order". As also mentioned previously, without a clear evidence for the long-ranged ferromagnetic order, interpretations using RSG phenomenon is misleading.

2. Analysis of the C K-edge XAS measurement needs significant improvement. HOPG is chemically very different material with very different electronic structure. Comparison should be made against coordination solids with similar chemical and electronic structure. In addition, references are needed when assigning the peaks. The authors also need to provide explanations for why the relative intensities and energy of the observed features remain similar between all Cu-ABDC materials while the magnetism and electronic structures are very different.

Reviewer #2:

Remarks to the Author:

The authors have made proper revisions and addressed all my concerns and question. Thus, the acceptance is recommended.

Response to Reviewers and a Summary of Changes

Many thanks to the reviewers for having given us valuable comments on the manuscript of NCOMMS-22-46092A submitted to the **Nat. Commun.**:

Title: Intrinsic room-temperature ferromagnetism in a two-dimensional semiconducting metal-organic framework

Authors: Sihua Feng, Hengli Duan, Hao Tan, Fengchun Hu, Chaocheng Liu, Yao Wang, Zhi Li, Liang Cai, Yuyang Cao, Chao Wang, Zeming Qi, Li Song, Xuguang Liu, Zhihu Sun and Wensheng Yan

Response to the Reviewers

Reply to Reviewer 1

Comment: The authors have mostly addressed the issues with additional experimental investigation and discussions. The manuscript has been much improved. However, a few important points remain.

Reply: Thank you very much for your positive comments. We have revised the manuscript according to your advice and hope it could be published in *Nat. Commun.*.

1. Question: The updated manuscript still lacks experimental evidences to support long-ranged ferromagnetic order in Cu-ABDC MOFs. The AC magnetic data exhibit broad in-phase susceptibility peak and the absence of out-of-phase susceptibility peak. Further, magnetic scattering from neutron diffraction data could not be observed. In fact, computational results stand as the single support for the long-ranged ferromagnetic order. The authors should clearly indicate this limitation in their manuscript by including any statements relevant to the following: “Computational results suggests long-ranged ferromagnetic order, but further experimental investigation are needed”. Specifically, while terms like “ferromagnetic coupling” may be fine, refrain from using the terms related to “long-ranged” and “magnetic order”. As also mentioned previously, without a clear evidence for the long-ranged ferromagnetic order, interpretations using RSG phenomenon is misleading.

Reply: Thank you for your very professional suggestions. Following the suggestions, we have performed neutron diffraction measurements at the energy-resolving neutron imaging spectrometer (ERNI) of the China Spallation Neutron Source (CSNS) and some magnetic susceptibility tests, to figure out whether the long-ranged magnetic order exists in our sample or not. After comprehensively reviewing all the magnetism-related data, we think that the experimental results strongly imply the existence of the long-range magnetic order. Additionally, angle-dependent C *K*-edge XAS measurements (**Supplementary Figure 25d**) also confirm that delocalized π electrons are present. In order to express more rigorously, we agree with the reviewer to weaken the statement about long-range magnetic order and focus on the local ferromagnetic coupling in the revised manuscript, which will not affect the main conclusion and theme of this article. We will briefly discuss the related experiment results in the following:

(1) Distinct hysteresis loops were observed for 1.4%- and 2.7%-LS-Cu-ABDC MOFs in the temperature range from 5 to 300 K, as shown in **Supplementary Figure 20b-c**. The remanent magnetization (M_r) is about $0.06 \mu_B/\text{Cu}$ with a coercive field of about 170 Oe at 5 K, and $0.02 \mu_B/\text{Cu}$ with a coercive field of 60 Oe at 300 K for 1.4%-Cu-ABDC MOF. For 2.7%-Cu-ABDC MOF, the M_r is about $0.01 \mu_B/\text{Cu}$ and $0.003 \mu_B/\text{Cu}$ at 5 K and 300 K, with a coercive field of about 100 Oe and 35 Oe, respectively. The relatively high remanent magnetization (about 12% at 5 K for 1.4%-Cu-ABDC MOF) at finite temperature indicates the presence of spontaneous magnetization, and hence the ferromagnetic order. The hysteresis loop of Cu-ABDC at 5 K in **Supplementary Figure 20a** is possibly due to the spin canting effect or defect [*Zhang et al., J. Appl. Phys.* **2000**, *87*, 6007; *Shen et al., J. Am. Chem. Soc.* **2012**, *134*, 17286-17290].

Supplementary Figure 20. The M - H curves at different temperatures for Cu-MOFs. M - H curves of Cu-ABDC (a), 1.4%-LS-Cu-ABDC (b) and 2.7%-LS-Cu-ABDC (c), respectively.

(2) We have also extended the temperature for temperature-dependent DC magnetic susceptibility and M - H measurements to 400 K, as shown in **Fig. R1**. Hysteresis loop is also observed at up to 400 K. Usually, hysteresis loop at such high temperature suggests the existence of FM order. The samples will decompose up to about 150 °C according to the thermogravimetric analysis (TGA) results [Murray *et al.*, *J. Am. Chem. Soc.* **2010**, *132*, 7856-7857; Zhan *et al.*, *Adv. Funct. Mater.* **2019**, *29*, 1806720], which hinders us from performing magnetic measurements at higher temperatures.

Fig. R1. (a) The M - T curves of 1.4%-Cu-ABDC MOF from 2 to 400 K. (b) M - H curve at 400 K.

(3) The neutron diffraction measurements were performed at 300 K in the d -spacing range of 0.5-9 Å. The intensity of the magnetic contribution to the diffraction peaks will decrease towards higher angles, thus, magnetic scattering peaks are most likely present in lower angles [Zhou *et al.*, *Phys. Rev. Lett.* **2011**, *106*, 147204; Sharma *et al.*, *J. Phys. Chem. Sol.* **2017**, *100*, 14-18]. We have obtained the neutron diffraction patterns of the pristine antiferromagnetic Cu-ABDC and ferromagnetic 1.4%-LS-Cu-ABDC samples. Since the magnetic interactions in our samples are relatively small, the intricate structure and the abundance of hydrogen atoms in our MOFS, prevent it from accurately matching our calculated crystal structure. However, compared to that of the pristine antiferromagnetic Cu-ABDC (**Supplementary Figure 27**), the neutron diffraction patterns of 1.4%-LS-Cu-ABDC exhibit a clear decrease at about 8.5 Å with the emergence of the intrinsic ferromagnetism, indicating the contribution of long-range magnetic order [Sibille *et al.*, *Phys. Rev. B* **2014**, *89*, 104413; Hagihala *et al.*, *npj Quantum Mater.* **2019**, *4*, 14; Viswanathan *et al.*, *J. Phys. Chem. C* **2019**, *123*, 18551-18559; Kozlenko *et al.*, *npj Quantum Mater.* **2021**, *6*, 19; Peng *et al.*, *Adv. Funct. Mater.* **2022**, *32*, 2106592]. However, due to the limitations of instruments and samples, it is difficult to carry out variable temperature tests.

Figure S27. Neutron diffraction patterns for Cu-ABDC and 1.4%-LS-Cu-ABDC MOFs.

(4) At last, we would like to discuss the AC magnetic susceptibility results (**Supplementary Fig. 21**). We fully understand the reviewer's concern about the AC magnetic susceptibility and rechecked the data. We think that no magnetic order-related AC magnetic susceptibility features being observed may be explained by that the test temperature is well below the transition temperature T_C or the response of our samples to AC magnetic susceptibility is relatively weak due to the weak magnetic moments. In our one parallel unpublished work on Co-BDC MOF with a similar organic linker of 1,4-benzenedicarboxylic acid (BDC) (**Fig. R2a**), which exhibits clear and lower transition temperature T_N , the clear frequency-independent AC magnetic susceptibility of the real part $\chi'(T)$ were detected. As shown in **Fig. R2b**, the ZFC magnetization values decrease with the temperature from about 43 K, which is consistent with conventional antiferromagnetic material. For the AC magnetic susceptibility real part in **Fig. R2c**, we observed that the real part $\chi'(T)$ curves exhibit a clear peak at about 43 K, and the location of this peak is independent of the frequency of the AC susceptibility measurement, a feature that is present in antiferromagnetic or ferromagnetic long-range order system [Singh *et al.*, *Phys. Rev. B* **2008**, 77, 144403; Bastien *et al.*, *Phys. Rev. B* **2019**, 99, 214410; Syzranov *et al.*, *Nat. Commun.* **2022**, 13, 2993]. Thus, it indicates the long-range magnetic order could be mediated by aromatic carboxylic acid linkers [Shen *et al.*, *J. Am. Chem. Soc.* **2012**, 134, 17286-17290; Zhou *et al.*, *Phys. Rev. Mater.* **2021**, 5, 074405].

Supplementary Figure 21. AC magnetic susceptibility measurement of 1.4%- and 2.7%-LS-Cu-ABDC MOF. (a-b) Temperature variation of the real part of the ac susceptibility measurement (a) and imaginary part (b) at various frequencies from 1 Hz to 1000 Hz with $H_{ac} = 2$ Oe and $H_{dc} = 0$ Oe from 50 to 360 K for 1.4%-LS-Cu-ABDC MOF. (c-d) Temperature variation of the real part of the ac susceptibility measurement (c) and imaginary part (d) at various frequencies from 1 Hz to 1000 Hz with $H_{ac} = 2$ Oe and $H_{dc} = 0$ Oe from 50 to 360 K for 2.7%-LS-Cu-ABDC MOF.

Fig. R2. (a) Structure model of Co-BDC MOF. (b) The ZFC curves of our unpublished Co-BDC MOF. (c) Temperature variation of the real part of the AC magnetic susceptibility measurement at various frequencies from 1 Hz to 1000 Hz with $H_{ac} = 2$ Oe and $H_{dc} = 0$ Oe from 2 to 100 K.

Accordingly, the description, “*However, we did not observe any frequency-dependent peaks in the AC magnetic susceptibility characterizations (see details in Supplementary Fig. 21), which are often attributed to short-range behavior such as*

spin glass. Meanwhile, AC magnetic susceptibility signals related to long-range order were also missing in the measurement, probably due to the test temperature being well below the transition temperature T_c or the response of our samples to AC magnetic susceptibility is relatively weak because of the weak magnetism.” and “*Angle-dependent C K-edge XAS (see details in Supplementary Figs.25) also confirms that delocalized π electrons are present. Furthermore, computational results suggest long-range order in our sample, which may cause the changed neutron diffraction patterns after ligand cleavage (see details in Supplementary Figs.26-27).*” **has been added in** pages 10 and 13 in the main text, and the origin description in page 10 in the main text, “*In addition, we note that the M-T curves of our samples are obviously different from that of a pure spin glass, which show a small but distinct maximum at a temperature of about 275 K. Above this temperature, M-H curves also show hysteresis. On the other hand, as the temperature decreases, a clear bifurcation of FC and ZFC curves exist at a lower temperature, namely thermomagnetic irreversibilities (TMI). Both the maximum in M-T and onset of TMI at lower temperature are the hallmarks of reentrant spin-glass (RSG) behavior. In the RSG state, long-range magnetic order appears in certain T regime, but the competition between FM and AFM leads to the partial or total breakdown of the high-temperature FM state to RSG state at the lower temperature, which is considered to be a gradual phase transformation. The spin configuration of this lower temperature RSG state consists of individual spins or small spin clusters frozen randomly with a trace of long-range FM order along the direction of the applied magnetic field.*” **has been changed to** “*In addition, we note that the magnetization of our samples increases from about 70 K to a local maximum at 270 K, in agreement with the competition between FM and AFM interactions, which has been studied as cooperative magnetism in complexes based on Cu dimer in the past.^{61,62}*”

2. Question: Analysis of the C K-edge XAS measurement needs significant improvement. HOPG is chemically very different material with very different electronic structure. Comparison should be made against coordination solids with similar chemical and electronic structure. In addition, references are needed when assigning the peaks. The authors also need to provide explanations for why the relative intensities and energy of the observed features remain similar between all Cu-ABDC materials while the magnetism and electronic structures are very different.

Reply: Thank you for your suggestions. According to your suggestions, we carried out the angle-dependent C K-edge XAS measurement with linear-polarized synchrotron

radiation X-ray to distinguish orbitals with two different symmetries, π and σ , in Cu-MOF and confirmed the existence of orbitals with π . The results are summarized below:

(1) The relative intensities and energy of the observed features remain similar between all Cu-ABDC materials while their magnetism and electronic structures are very different. The reason is that the C *K*-edge XAS represents the transition from C 1*s* to unoccupied molecular orbitals with C 2*p* components (**Supplementary Figure 25a**), and the magnetism of Cu-MOF is mainly related to the hybrid between Cu 3*d* orbitals. C *K*-edge XAS depends directly on the local coordination environment of C atoms, and the cleaving strategy mainly alters the local coordinated environment of the Cu atoms, resulting in slight changes in the overall lattice structure of the MOF. Therefore, the relative intensities and energy of the observed features in C *K*-edge XAS remain similar between all Cu-ABDC materials.

(2) The C *K*-edge XAS spectra (**Supplementary Figure 25b**) of Cu-ABDC MOFs can be roughly divided into two regions according to the incident photon energy due to the random orientation of the samples. Energy region before 293 eV can be assigned to the π^* region, and in higher energy region above 293 eV can be attributed to σ^* region [*Liang et al., Nat. Mater.* **2011**, *10*, 780-786; *Wang et al., Energy Environ. Sci.* **2013**, *6*, 2900-2906; *Han et al., Nat. Commun.* **2020**, *11*, 2209; *Pan et al., Nature* **2023**, *614*, 95-101; *Liang et al., J. Am. Chem. Soc.* **2012**, *134*, 3517-3523; *Wang et al., J. Am. Chem. Soc.* **2017**, *139*, 9419-9422; *Hu et al., Nat. Commun.* **2021**, *12*, 1854]. Specifically, the photon energy position of π^* in our samples indicates electronic transitions from C 1*s* core level to delocalized π^* energy level in Cu-ABDC MOFs, which suggests the delocalized π electrons are present in our MOFs.

(3) To obtain the angle-dependent C *K*-edge XAS spectra, the Cu-MOF was carefully dissolved in ethanol and spin-coated on a silicon wafer to fabricate an oriented-aggregated sample. Due to the π - π stacking interaction, the Cu-MOF sheets prefer to be deposited on silicon wafer with the benzene ring roughly perpendicular to the substrate. In other words, the aromatic ring of Cu-MOF is along the normal of the sample. Hence, we can utilize the linear polarized X-ray to district the orbitals with π and σ symmetry (**Supplementary Figure 25c**). The region before 293 eV where the intensity increases with the angle of incident light can be assigned to π^* orbitals (**Supplementary Figure 25d**). The highest intensity was obtained when the angle between the electric field vector *E* of the incident light and the normal of the sample is 90°, indicating that the final states are the orbitals perpendicular to the aromatic ring plane, e.g. the π orbitals. In higher energy σ^* region above 293 eV, the intensity decreases with the angle of

incident light. Here, the final state of transition is within the aromatic ring plane, which is with σ symmetry. In other similar systems such as graphene oxide and cobalt phthalocyanine, π and σ orbitals are also assigned at similar energies [Gandhiraman *et al.*, *J. Phys. Chem. C* **2014**, *118*, 18706-18712; Basagni *et al.*, *ACS Nano* **2016**, *10*, 2644-2654; Gregory *et al.*, *J. Phys. Chem. C* **2017**, *121*, 9142-9152; Son *et al.*, *J. Am. Chem. Soc.* **2016**, *138*, 8096-8103].

Figure S25. (a) Schematic illustration of an X-ray adsorption process of a diatomic molecule. (b) C *K*-edge XAS of Cu-ABDC MOFs. (c) The schematic of π and σ bond. (d) Angle-dependent XAS spectra of 1.4%-Cu-ABDC MOF at C *K*-edge. The measurement geometry is shown as an insert.

Accordingly, the origin description in page 28 in the Supplementary Information, “The C *K*-edge XAS spectra of Cu-ABDC MOFs exhibit four distinct absorption peaks. A domain peak A at about 285.3 eV is attributed to C-C π^* (ring) excitations, and peak B (286.4 eV), peak C (287.7 eV) and peak D (288.9 eV) can be ascribed to the functional groups containing O and N. Specifically, the photon energy position of π^* in our samples is in accordance with that of HOPG, indicating electronic transitions from C 1s core level to delocalized π^* energy level in Cu-ABDC MOFs. Therefore, we can conclude that delocalized π electrons are present in our MOFs, and the long-range ferromagnetic order is preferred due to the coupling between localized spin states via delocalized π electrons in the organic linkers.” has been changed to “The relative

intensities and energy of the observed features remain similar between all Cu-ABDC materials while their magnetism and electronic structures are very different. The reason is that the C K-edge XAS represents the transition from C 1s to unoccupied molecular orbitals with C 2p components (Figure S25a), and the magnetism of Cu-MOF is mainly related to the hybrid between Cu 3d orbitals. C K-edge XAS depends directly on the local coordination environment of C atoms, and the cleaving strategy mainly alters the local coordinated environment of the Cu atoms, resulting in slight changes in the overall lattice structure of the MOF. Therefore, the relative intensities and energy of the observed features in C K-edge XAS remain similar between all Cu-ABDC materials. The C K-edge XAS spectra (Figure S25b) of Cu-ABDC MOFs exhibit two regions according to the incident photon energy. Due to the random orientation of the sample, before 293 eV can be assigned to the π^* region, and in higher energy regions above 293 eV can be attributed to σ^* region. Specifically, the photon energy position of π^* in our samples indicates electronic transitions from C 1s core level to delocalized π^* energy level in Cu-ABDC MOFs, which suggests the delocalized π electrons are present in our MOFs.

To obtain the angle-dependent C K-edge XAS spectra, the Cu-MOF was carefully dissolved in ethanol and spin-coated on a silicon wafer to fabricate an oriented-aggregated sample. Due to the π - π stacking interaction, the Cu-MOF sheets prefer to be deposited on silicon wafer with the benzene ring roughly perpendicular to the substrate. In other words, the aromatic ring of Cu-MOF is along the normal of the sample. Hence, we can utilize the linear polarized X-ray to distinguish the orbitals with π and σ symmetry (Figure S25c). The region before 293 eV where the intensity increases with the angle of incident light can be assigned to π^* orbitals (Figure S25d). The highest intensity was obtained when the angle between the electric field vector E of the incident light and the normal of the sample is 90° , indicating that the final states are the orbitals perpendicular to the aromatic ring plane, e.g. the π orbitals. In higher energy σ^* region above 293 eV, the intensity decreases with the angle of incident light. Here, the final state of transition is within the aromatic ring plane, which is with σ symmetry. In other similar systems such as graphene oxide and cobalt phthalocyanine, π and σ orbitals are also assigned at similar energies.”

Reply to Reviewer 2

Comment: The authors have made proper revisions and addressed all my concerns and questions. Thus, the acceptance is recommended.

Reply: Thank you very much for your support in publishing our work in *Nat. Commun.*.

Reviewers' Comments:

Reviewer #1:

Remarks to the Author:

The authors have made proper revisions, and so the acceptance is recommended.

Response to Reviewers' Comments

We are grateful to reviewer #1 for the comment on the manuscript of NCOMMS-22-46092B submitted to the journal of *Nature Communications*:

Title: Intrinsic room-temperature ferromagnetism in a two-dimensional semiconducting metal-organic framework

Authors: Sihua Feng, Hengli Duan, Hao Tan, Fengchun Hu, Chaocheng Liu, Yao Wang, Zhi Li, Liang Cai, Yuyang Cao, Chao Wang, Zeming Qi, Li Song, Xuguang Liu, Zhihu Sun and Wensheng Yan

Reviewer #1 (Remarks to the Author)

The authors have made proper revisions, and so the acceptance is recommended.

We sincerely appreciate the reviewer's recommendation of our manuscript for publication. We also thank the reviewer for the affirmation and appreciation of our efforts. It is our pleasure that the reviewer accepts our work that a 2D semiconducting antiferromagnetic Cu-MOF can be endowed with intrinsic room-temperature ferromagnetic coupling using a ligand cleavage strategy to regulate the inner magnetic interaction within the Cu dimers, which may pave the way toward design and fabricate MOF-based semiconducting room-temperature ferromagnetic materials and promotes their practical applications in next-generation spintronic devices.